# Faster Diffusion via Temporal Attention Decomposition

**Haozhe Liu** [1,4,✉]**, Wentian Zhang** [†]**, Jinheng Xie** [2,†]**, Francesco Faccio** [1,3]**, Mengmeng Xu** [4]**, Tao Xiang** [4]**, Mike Zheng Shou** [2]**, Juan-Manuel Perez-Rua** [4]**, Jürgen Schmidhuber**[1,3]

[1]*Center of Excellence for Generative AI, King Abdullah University of Science and Technology (KAUST)*

[2] *Show Lab, National University of Singapore (NUS)*

[3] *Swiss AI Lab, IDSIA, USI & SUPSI, Lugano* [4] *Meta AI*

**Reviewed on OpenReview:** *https://openreview.net/forum?id=xXs2GKXPnH*

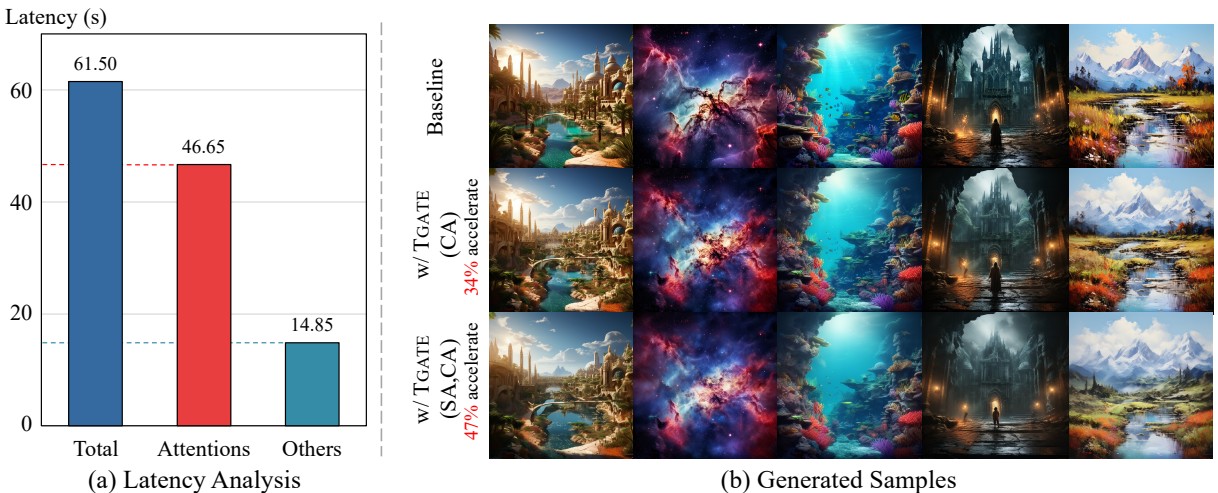

(a) Latency Analysis          (b) Generated Samples

Figure 1: While generating a 1024×1024 image over 25 steps using a well-known text-to-image diffusion model (Chen et al., 2024b), we analyze the latency contributions of its various components. The major computational bottleneck is the attention mechanism, which exhibits a key pattern during inference: cross-attention is initially essential, but its significance decreases over time. Conversely, self-attention has minimal initial impact but becomes crucial over time. This allows for caching attention maps and reusing them when they are less crucial, thereby considerably speeding up inference with slight impact on generation quality, as illustrated in (b).

## Abstract

We explore the role of attention mechanism during inference in text-conditional diffusion models. Empirical observations suggest that cross-attention outputs converge to a fixed point after several inference steps. The convergence time naturally divides the entire inference process into two phases: an initial phase for planning text-oriented visual semantics, which are then translated into images in a subsequent fidelity-improving phase. Cross-attention is essential in the initial phase but almost irrelevant thereafter. However, self-attention initially plays a minor role but becomes crucial in the second phase. These findings yield a simple and training-free method known as temporally gating the attention (TGATE), which efficiently generates images by caching and reusing attention outputs at scheduled time steps. Experimental results show when widely applied to various existing text-conditional diffusion models, TGATE accelerates these models by 10%–50%. The code of TGATE is available at https://github.com/HaozheLiu-ST/T-GATE.

---

*Corresponding Author: haozhe.liu@kaust.edu.sa      [†]Equal Contribution

# 1 Introduction

" A small leak will sink a great ship."

—Benjamin Franklin

Diffusion models (Jarzynski, 1997; Neal, 2001; Ho et al., 2020) have been widely used for image generation. Featuring an **attention mechanism** (Vaswani et al., 2017; Schmidhuber, 1992b), they align different modalities (Rombach et al., 2022), including text, to generate high-quality images and videos. Several studies highlight the importance of attention for spatial control (Xie et al., 2023; Hertz et al., 2023; Chefer et al., 2023); however, only a few have investigated its role from a temporal perspective during denoising.

The attention module generally comprises self-attention, which processes the context across spatial positions, and cross-attention, which integrates signals from various modalities. Understanding the roles of these components during inference is key to comprehend the overall model behavior. We analyze their contributions at different time steps and gained three critical insights:

- **Cross-attention outputs convergence to a fixed point in first several steps.** Accordingly, the time point of convergence divides the denoising of diffusion models into two phases: i) an initial phase, during which the model relies on cross-attention to plan text-oriented visual semantics; this is denoted as the *semantic-planning* phase, and ii) a subsequent phase, during which the model learns to generate images from previous semantic planning; this is referred to as the *fidelity-improving phase*.

- **Cross-attention is redundant in the fidelity-improving phase.** During the semantics-planning phase, cross-attention plays a crucial role in creating meaningful semantics. However, in the latter phase, it converges and has a minor impact on image generation. Bypassing cross-attention during the fidelity-improving phase can indeed potentially reduce computational costs while maintaining the image generation quality.

- **Self-attention is largely redundant in the semantics-planning phase.** Unlike cross-attention, self-attention evidently plays a significant role in the later phase. However, its contribution is limited in the early semantics-planning phase. By selectively skipping self-attention during this phase, the inference process can be further accelerated with only minor impact on generation.

Notably, the scaled dot product in the attention mechanism is a quadratic complexity operation. As the resolution and token length in modern models increase, attention mechanism inevitably increases computational costs and becomes a significant source of latency (Li et al., 2024). Thus, the role of attention mechanism must be re-evaluated; moreover, the aforementioned shortcoming inspires us to design a simple, effective, and training-free method, *i.e.*, **t**emporally **gat**ing the attention (**TGATE**), to improve the efficiency and maintain the quality of images generated by off-the-shelf diffusion models. The principal observations with respect to TGATE are as follows:

- TGATE increases efficiency by caching and reusing the cross-attention outcomes when they are rendered useless, thereby eliminating the calculation of redundant attention. This strategy does not affect the model performance, as the predictions of cross-attention converge and are potentially redundant.

- TGATE is training-free and has broad applicability in text-to-image and text-to-video models, and supports U-Net and transformer-based architectures. It is also orthogonal to different noise schedulers and acceleration methods.

- TGATE can further accelerate diffusion models by dynamically caching and reusing self-attention predictions during the initial phase. In extreme cases, TGATE reduces the multiply-accumulate (MAC) operation of PixArt-Alpha from 107 T to 64 T and cuts its latency from 62 s to 33 s on a 1080Ti commercial card, thereby enhancing the efficiency without considerably impacting the performance.

# 2 Preliminary

Diffusion technique has a rich history, dating back to nonequilibrium statistical physics (Jarzynski, 1997) and annealed importance sampling (Neal, 2001). This mechanism characterized by its scalability and stability, (Dhariwal & Nichol,

2021; Ho et al., 2020), has been widely used in modern text-conditional generative models (Rombach et al., 2022; Ramesh et al., 2022; 2021; Saharia et al., 2022; Chen et al., 2024c; Brooks et al., 2024).

**Learning Objective.** Herein, the formulation introduced by LDM (Rombach et al., 2022) is used to construct a latent diffusion model comprising four main components: an image encoder $\mathcal{E}(\cdot)$, an image decoder $\mathcal{D}(\cdot)$, a denoising model $\epsilon_\theta(\cdot)$, and a text embedding $c$. The learning objective for this model is defined as follows:

$$\mathcal{L}_\theta = \mathbb{E}_{z_0 \sim \mathcal{E}(x), t, c, \epsilon \sim \mathcal{N}(0,1)} \left[ ||\epsilon - \epsilon_\theta(z_t, t, c)||_2^2 \right], \tag{1}$$

where $\epsilon_\theta(\cdot)$ is designed to accurately estimate noise $\epsilon$ added to the current latent representation $z_t$ at time step $t \in [1, n]$, that is conditioned on text embedding $c$. During inference, $\epsilon_\theta(z_t, t, c)$ is called multiple times to recover $z_0$ from $z_t$, where $z_0$ is decoded into an image $x$ using $\mathcal{D}(z_0)$.

**Inference Stage.** In this stage, classifier-free guidance (CFG) (Ho & Salimans, 2022) is commonly employed to incorporate conditional guidance as follows:

$$\epsilon_{c,\theta}(z_t, t, w, c) = \epsilon_\theta(z_t, t, \varnothing) + w(\epsilon_\theta(z_t, t, c) - \epsilon_\theta(z_t, t, \varnothing)), \tag{2}$$

where $\varnothing$ represents the embedding of a null text, *i.e.*, "", $w$ is the guidance scale parameter, and $\epsilon_{c,\theta}$ implicitly estimates $p(c|z) \propto p(z|c)/p(z)$ to guide conditional generation $\tilde{p}(z|c) \propto p(z|c)p^w(c|z)$. In particular, $\nabla \log(p(c|z)) \propto \nabla_z \log(p(z|c)) - \nabla_z \log(p(z))$, which is identical to Eq. 2.

**Attention Mechanism.** In the denoising model $\epsilon_\theta$, each block extensively integrates the attention mechanism. Specifically, the self-attention module captures context across spatial positions, whereas the cross-attention module enables interactions with various input modalities, including text. The attention process is mathematically defined as follows:

$$\mathbf{C}_c^t = \text{Softmax}\left(\frac{Q_z^t \cdot K}{\sqrt{d}}\right) \cdot V, \tag{3}$$

where $Q_z^t$ represents a projection of $z_t$. For cross-attention, $K$ and $V$ are projections of the text embedding $c$. However, in self-attention, they are derived from $z_t$. $d$ denotes the feature dimension of $K$. This mechanism can be understood as querying learnable key-value codes, where each token's prediction is derived from a weighted sum of all values ($V$). The weights are determined by the similarity between the query ($Q$) and the corresponding keys ($K$) Despite its effectiveness, the attention mechanism acts as a significant computational bottleneck when processing high-resolution input due to its quadratic computational complexity.

## 3 Temporal Analysis of Attention Mechanism

Here, the role and functionality of attention mechanism in the inference stage of a well-trained diffusion model are discussed. First, an empirical observation of cross-attention map convergence is discussed in Section 3.1, followed by a systematic analysis of this observation in Section 3.2. Section 3.3 concludes with a follow-up analysis of self-attention.

### 3.1 Convergence of Cross-Attention Map

Cross-attention mechanisms provide textual guidance at each step in diffusion models. However, the shifts in the noise input across these steps pose this question: *Do the feature maps generated by cross-attention exhibit temporal stability, or do they fluctuate over time?*

To find an answer, we randomly collect 1,000 captions from the MS-COCO dataset and generate images using a

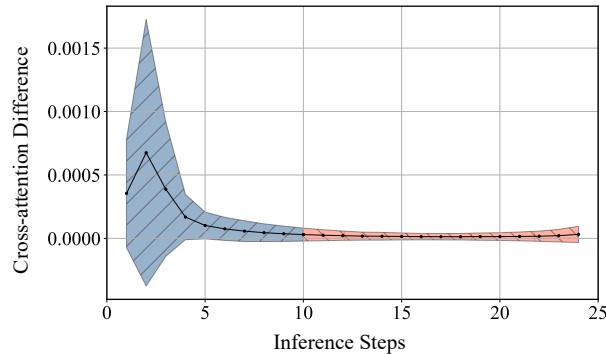

Figure 2: **Difference in cross-attention maps between two consecutive inference steps on the MS-COCO dataset.** Each data point in the figure is an average of 1,000 captions and all cross-attention maps within the model. The shaded area indicates the variance, whereas the curve demonstrates that the difference between consecutive steps progressively approaches zero.

pre-trained SD-2.1 model[1] which is based on DPM solver
(Lu et al., 2022) with 25 inference steps. During inference,
we calculate the L2 distance between $\mathbf{C}^t$ and $\mathbf{C}^{t+1}$, where $\mathbf{C}^t$ represents the cross-attention maps at time step $t$. The difference in cross-attention between the two steps is calculated by averaging L2 distances among all input captions, conditions, and depths.

Fig. 2 shows the variation in cross-attention differences across various inference steps. A clear trend is visible, showing a gradual convergence of differences toward zero. Convergence always appears within 5-10 inference steps. Therefore, cross-attention maps converge to a fixed point and do not offer dynamic guidance for image generation. This finding supports the effectiveness of CFG with respect to cross-attention, demonstrating that despite varying conditions and initial noise, unconditional and conditional batches can converge toward a single, consistent result (Castillo et al., 2023). We also track the cross-attention differences across different blocks; refer to Appendix E for details.

This phenomenon shows that the impact of cross-attention during inference process is not uniform and inspires the temporal analysis of cross-attention.

## 3.2   Role of Cross-Attention in Inference

**Analytical Tool.**  Existing analysis (Ma et al., 2024) shows that the consecutive inference steps of diffusion models have similar denoising behaviors. Inspired by behavioral explanation (Bau et al., 2020; Liu et al., 2023), we measure the impact of cross-attention by effectively "removing" it at a specific phase and observing the resulting difference in the image generation quality. In practice, this removal is approximated by substituting the original text embedding with a placeholder for a null text, *i.e.*, "". We formalize the standard denoising trajectory as a sequence as follows:

$$\mathbf{S} = \{\epsilon_{\text{c}}(z_n, c), \epsilon_{\text{c}}(z_{n-1}, c), ..., \epsilon_{\text{c}}(z_1, c)\}, \tag{4}$$

where we omit the time step $t$ and guidance scale $w$ for simplicity. The image generated from sequence $\mathbf{S}$ is denoted by $x$. This standard sequence is then modified by replacing the conditional text embedding $c$ with the null text embedding $\varnothing$ over a specified inference interval, resulting in two new sequences, $\mathbf{S}_m^{\text{F}}$ and $\mathbf{S}_m^{\text{L}}$, based on a scalar $m$ as follows:

$$\mathbf{S}_m^{\text{F}} = \{\epsilon_{\text{c}}(z_n, c), \cdots, \epsilon_{\text{c}}(z_m, c), \cdots, \epsilon_{\text{c}}(z_1, \varnothing)\},$$
$$\mathbf{S}_m^{\text{L}} = \{\epsilon_{\text{c}}(z_n, \varnothing), \cdots, \epsilon_{\text{c}}(z_m, \varnothing), \cdots, \epsilon_{\text{c}}(z_1, c)\}. \tag{5}$$

Here, $m$ serves as a *gate step* that splits the trajectory into two phases. In sequence $\mathbf{S}_m^{\text{F}}$, the null text embedding $\varnothing$ replaces the original text embedding $c$ for the steps from $m + 1$ to $n$. In contrast, in sequence $\mathbf{S}_m^{\text{L}}$, the steps from 1 to $m$ use the null text embedding $\varnothing$ instead of the original text embedding $c$, whereas the steps from $m$ to $n$ continue to use the original text embedding $c$. The images generated from these two trajectories are denoted as $x_m^{\text{F}}$ and $x_m^{\text{L}}$, respectively. To determine the impact of cross-attention at different phases, the differences in generation quality among $x$, $x_m^{\text{L}}$, and $x_m^{\text{M}}$ are compared. If the image generation quality among $x$ and $x_m^{\text{F}}$, are considerably different, it indicates the importance of cross-attention at that phase. If the quality does not vary considerably, the inclusion of cross-attention may not be necessary.

Herein, SD-2.1 is used as the model, and the DPM solver (Lu et al., 2022) is used for noise scheduling. The total inference step in all experiments is set as 25. The text prompt, "High quality photo of an astronaut riding a horse in space." is used for visualization.

**Results and Discussions.**  Fig. 3(a) shows the trajectory of the mean of predicted noise, which empirically shows that denoising converges after 25 inference steps. Therefore, analyzing the impact of cross-attention within this interval is difficult. As shown in Fig. 3(b), the gate step $m$ is set to 10, which yields three trajectories: $\mathbf{S}$, $\mathbf{S}_m^{\text{F}}$ and $\mathbf{S}_m^{\text{L}}$. The visualization illustrates that ignoring the cross-attention after 10 steps does not influence the final outcome. However, a notable disparity is observed after bypassing cross-attention in the initial steps. As shown in Fig. 3(c), the image generation quality (Fréchet inception distance, FID) considerably deteriorates in the MS-COCO validation set due to this elimination. The resulting quality is even worse than the weak baseline that generates images without CFG. We then generalize these assessments to a range of gate steps, inference numbers, noise schedulers, and base models. The experimental results consistently show that the FIDs of $\mathbf{S}_m^{\text{F}}$ are slightly better than the baseline $\mathbf{S}$ and outperform

---

[1]https://huggingface.co/stabilityai/stable-diffusion-2-1

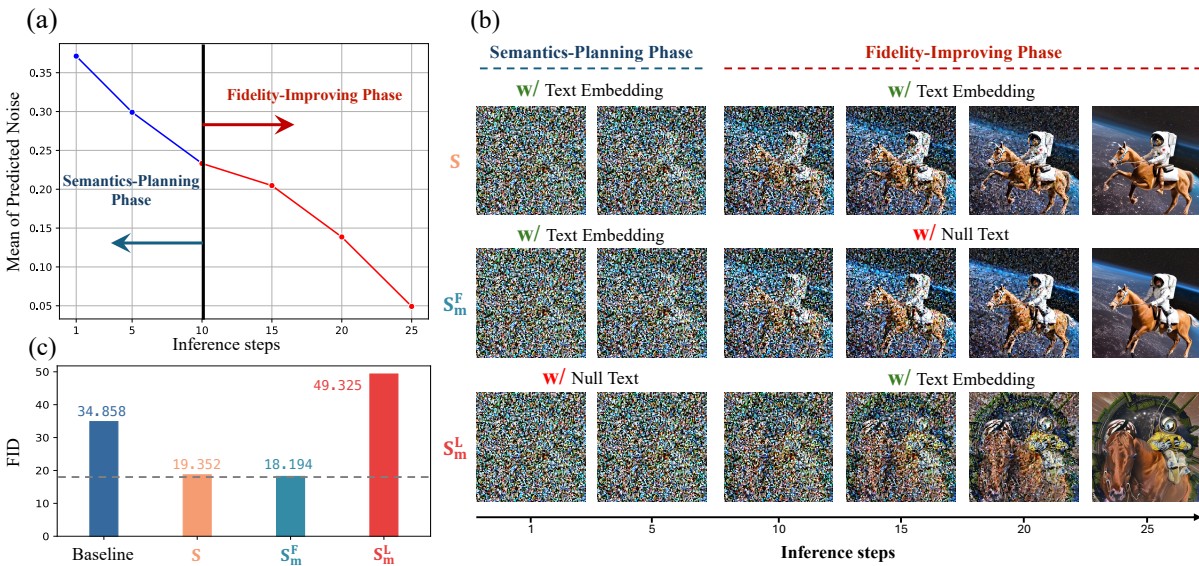

Figure 3: **Impact of cross-attention on the inference steps in a pre-trained diffusion model, *i.e.*, stable diffusion 2.1 (SD-2.1)**. (a) The mean of the noise predicted at each inference step. (b) Images generated by the diffusion model at different inference steps. The first row, $\mathbf{S}$, in (b) feeds text embedding to cross-attention modules for all steps, the second row, $\mathbf{S}_m^{\mathrm{F}}$, only uses text embedding from the first step to the $m$-th step, and the third row, $\mathbf{S}_m^{\mathrm{L}}$, inputs text embedding from the $m$-th to the $n$-th step. (c) Zero-shot FID scores based on these three settings on the MS-COCO validation set (Lin et al., 2014), with the baseline defined as conditional generation without CFG. Here, FID is calculated using the full COCO validation set (Lin et al., 2014).

$\mathbf{S}_m^{\mathrm{L}}$ by a wide margin. These empirical observations consistently underscore the broad applicability of the reported findings over different configurations. Appendix A details these results.

These analyses can be summarized as follows:

- Cross-attention converges early during inference, which can be characterized by semantics-planning and fidelity-improving phases. The impact of cross-attention is not uniform in these two phases.

- Cross-attention in the semantics-planning phase is significant for generating semantics aligned with the text conditions.

- The fidelity-improving phase mainly improves the image quality without requiring cross-attention. FID scores can be slightly improved via null-text embedding in this phase.

### 3.3 Role of Self-Attention in Inference

**Analytical Tool** Unlike cross-attention, the direct removal of self-attention during inference is not a straightforward process, as the performance deteriorates considerably. Thus, determining the specific contributions of each time step is challenging. To address this issue, a novel analytical approach involving caching and reusing features is proposed. As our core premise, if the features of self-attention can be cached and reused across multiple steps without performance decline, they may be considered less critical. Building on this concept and drawing parallels with cross-attention analysis, two distinct trajectories, $\mathbb{S}^F$ and $\mathbb{S}^L$, are introduced.

In $\mathbb{S}^F$, self-attention features are cached and reused in the fidelity-improving phase using a gate step $m$ and interval $k$. Specifically, after $m$ inference steps, self-attention is reused for $k$ steps, and self-attention prediction is updated once for the next $k$-step reuse cycle. Contrarily, $\mathbb{S}^L$ skips self-attention during the initial semantics-planning phase for every $k$ step after several warm-up steps, typically set at 2. Then, self-attention is fully integrated into inference after $m$ steps. The performance difference between $\mathbb{S}^F$ and $\mathbb{S}^L$ indicates the contribution of self-attention at different phases.

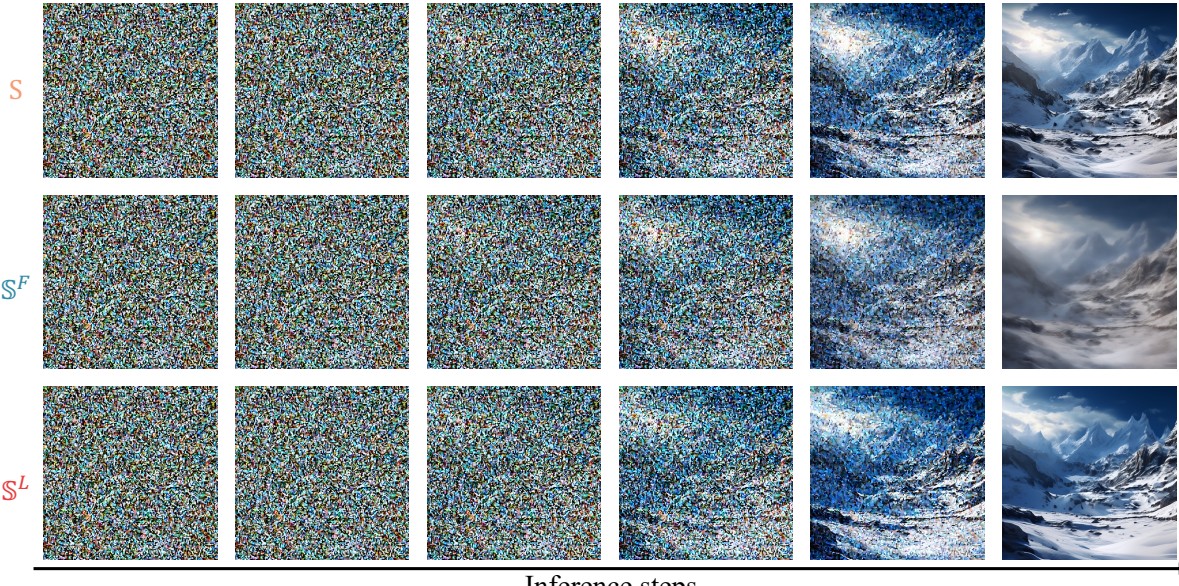

Figure 4: Illustration of the impact of self-attention on inference in Stable Diffusion 2.1 (SD-2.1). The base trajectory, $S$, does not use cached self-attention features. In $\mathbb{S}^F$, these features are cached and reused during the fidelity-improving phase. Conversely, $\mathbb{S}^L$ bypasses self-attention in the initial semantics-planning phase. The interval is set to 5 and the gate step to 10. The visualization shows that caching self-attention in the semantics-planning phase does not significantly affect the generation result. The input prompt is "paisaje montañoso nevado".

**Results and Discussions** As shown in Fig. 4, the visualization convincingly suggests that self-attention is more important in the latter inference phases. For quantitative analysis, various experiments are conducted using different values of gate step $m$ and interval $k$, which are detailed in Appendix B.

The observations from the analysis can be summarized as follows:

- Unlike cross-attention, bypassing self-attention increases FID scores, indicating quality degradation. However, selectively skipping it during the semantics-planning phase results in a minor, manageable performance drop.

- By increasing the interval for reusing the features, the efficiency can be improved but at the cost of performance, suggesting that it cannot be removed totally in the semantics-planning phase.

## 4 Proposed Method - TGATE

Results of the empirical study show that self-attention and cross-attention in the initial and last inference steps, respectively, are redundant. However, it is nontrivial to drop/replace attention modules without retraining the model. To this end, an effective and training-free method is proposed herein: TGATE. This method caches the attention outcomes and reuses them throughout the scheduled time steps.

### 4.1 Skipping Cross-Attention in the Fidelity-Improving Phase

**Caching Cross-Attention Maps.** Suppose $m$ is the gate step for the phase transition. In the $m$-th step and $i$-th cross-attention module, two cross-attention maps, $\mathbf{C}_c^{m,i}$ and $\mathbf{C}_\varnothing^{m,i}$, can be accessed from CFG-based inference. The average of these two maps is calculated to serve as an anchor and store it in a first-in-first-out feature cache $\mathbf{F}$. After traversing all the cross-attention blocks, $\mathbf{F}$ can be written as follows:

$$\mathbf{F} = \{\frac{1}{2}(\mathbf{C}_\varnothing^{m,i} + \mathbf{C}_c^{m,i})|i \in [1, l]\}, \tag{6}$$

where $l$ denotes the total number of cross-attention modules.

**Re-using Cached Cross-Attention Maps.** In each step of the fidelity-improving phase, when a cross-attention operation is performed during the forward pass, it is omitted from the computation graph. Instead, the cached $\mathbf{F}$[i] is fed into subsequent computations. This approach does not yield identical predictions at each step, as a residual connection (Hochreiter, 1991; Srivastava et al., 2015; He et al., 2016) in the neural networks allows the model to bypass cross-attention.

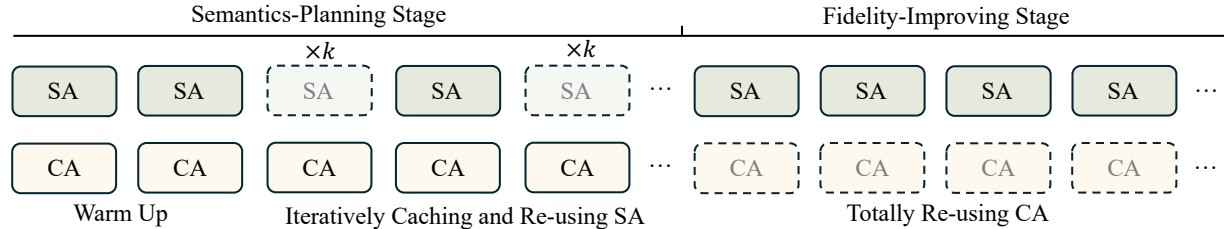

Figure 5: **Pipeline of TGATE for accelerating inference.** During the semantics-planning phase, cross-attention (CA) is continuously active, whereas self-attention (SA) is applied every $k$ steps following an initial warm-up period to conserve computational resources. In the fidelity-improving phase, cross-attention is substituted with a caching mechanism, and self-attention remains operational.

### 4.2 Skipping Self-Attention in Semantics-Planning Phase

The analysis described in Appendix B demonstrates that self-attention contributes mainly to the second phase, suggesting a reduction in its usage in the first phase. However, unlike cross-attention, self-attention cannot be entirely bypassed without considerably degrading the capacity and performance of the model. An interval caching strategy is introduced to preserve the generation performance. In particular, after activating self-attention with initial warm-up steps, output from all blocks is cached and reused for every $k$ step in the semantics-planning phase. In the fidelity-improving phase, self-attention is fully operational. The pipeline of TGATE is detailed in Fig. 5.

## 5 Related Works

Herein, the role and functionality of cross-attention within diffusion trajectories are analyzed. These factors have been previously studied from different perspectives. Spectral diffusion (Yang et al., 2023) traces diffusion trajectory via frequency analysis and finds that the diffusion model restores an image from varying frequency components at each step. T-stitch (Pan et al., 2024) shows that at the beginning of the inference, different models generated similar noise. This finding suggests that a smaller model can produce the same noise as a larger one, thereby considerably reducing the computational costs. By analyzing prompt switching effects, eDiff-I (Balaji et al., 2022) reveals that diffusion models respond to text prompts with distinct temporal dynamics, showing better comprehension of text signals in early denoising steps and diminishing in later ones. Adaptive guidance (Castillo et al., 2023) models diffusion trajectory as a graph and applies neural architecture search (NAS) to automatically identify the importance of each step. This approach identifies CFG (Ho & Salimans, 2022) as a redundant operator in some inference steps, suggesting the removal of unconditional batch for accelerating the generation speed. Building on similar foundations, recent studies (Kynkäänniemi et al., 2024; Sadat et al., 2023) suggest that strategically dropping or modifying the CFG scale can enhance generation performance. As per DeepCache (Ma et al., 2024), predictions from each block contain temporal similarities in consecutive time steps. Thus, reutilizing predictions from these blocks can improve the efficiency of the inference. Wimbauer et al. (Wimbauer et al., 2023) propose a contemporary work for caching block features; however, it requires a resource-friendly training process.

To the best of our knowledge, this study is orthogonal to the existing studies. We observe that cross-attention and self-attention non-uniformly yet independently (almost complementarily) contribute to the final generated samples across various time steps. Therefore, attention outcomes can be selectively copied and reused in certain inference steps without affecting the generation performance. This may inspire several new studies toward developing faster diffusion models.

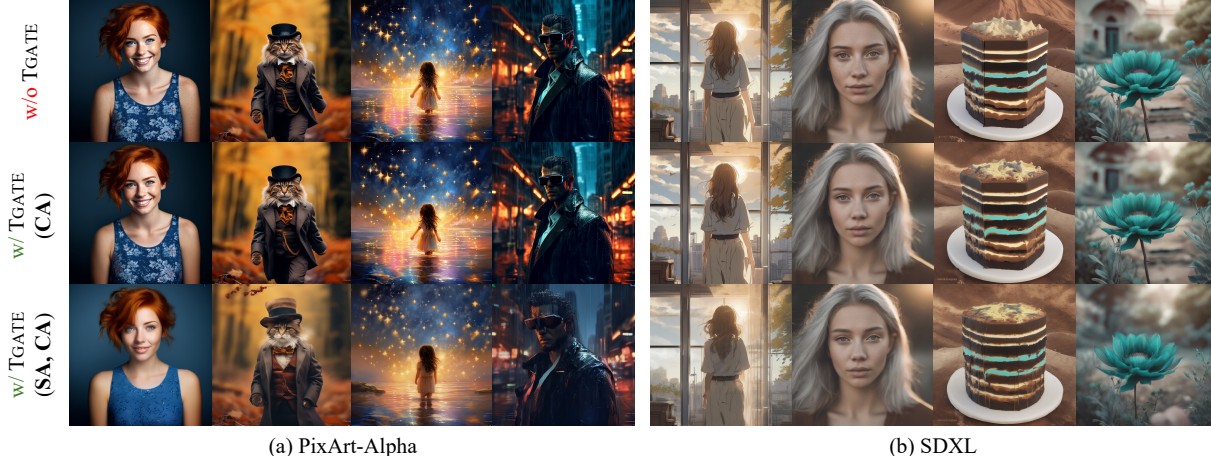

(a) PixArt-Alpha                                        (b) SDXL

Figure 6: **Samples generated from (a) PixArt-Alpha and (b) SDXL with or without TGATE given the same initial noise and captions.** The visualization of PixArt is generated using two configurations, i.e., $m$=15 or ($m$=15,$k$=5). For SDXL, the configuration is $m$=10 and ($m$=10,$k$=5). Refer to Appendix E for visualizations with different hyper-parameters ($m$ and $k$).

## 6  Experimental Results

The proposed method is integrated into several state-of-the-art diffusion models, including SD-series (Rombach et al., 2022; Podell et al., 2023; Blattmann et al., 2023), PixArt (Chen et al., 2024b), and OpenSora (Lab & etc., 2024). Following established evaluation protocols (Podell et al., 2023; Li et al., 2023; Lab & etc., 2024), comprehensive experiments are conducted using the MS-COCO (Lin et al., 2014), MJHQ (Li et al., 2023), OpenSora-Sample (Lab & etc., 2024) and DPG-Bench (Hu et al., 2024) datasets. The inference configuration, including the number of inference steps and the noise scheduler, follows the default settings for each model. Additionally, the proposed method is compared with other accelerating methods, such as the latent consistency model (Luo et al., 2023), adaptive guidance (Castillo et al., 2023), and DeepCache (Ma et al., 2024). Latency and MACs are considered the metrics for efficiency. MACs stand for Multiply–Accumulate Operations per image, which is automatically generated using Calflops (xiaoju ye, 2023). The generation performance is evaluated using FID (Heusel et al., 2017), CLIP score (Radford et al., 2021), and DPG-Score (Hu et al., 2024). More details are given in Appendix C.

Table 1: Computational complexity, latency, and FID on the MJHQ-10K (Li et al., 2023) using the base model of PixArt-Alpha (Chen et al., 2024b). The latency of generating one image is tested on a 1080 Ti commercial card. FID (Heusel et al., 2017) is used to test the performance of PixArt and CLIP score (Radford et al., 2021) for OpenSora (Lab & etc., 2024).

| Inference Method | Caching Modules | MACs | Latency | Generation Performance (FID) |
|---|---|---|---|---|
| PixArt-Alpha | - | 107.031T | 61.502s | 9.653 |
| PixArt-Alpha + TGATE ($m$=10) | CA | 70.225T | 40.648s | 11.268 |
| PixArt-Alpha + TGATE ($m$=15) | CA | 82.494T | 47.599s | **9.548** |
| PixArt-Alpha + TGATE ($m$=10, $k$=3) | CA,SA | 65.355T | 34.391s | 11.789 |
| PixArt-Alpha + TGATE ($m$=10, $k$=5) | CA,SA | **64.138T** | **32.827s** | 12.738 |
| PixArt-Alpha + TGATE ($m$=15, $k$=3) | CA,SA | 73.971T | 36.650s | 10.289 |
| PixArt-Alpha + TGATE ($m$=15, $k$=5) | CA,SA | 71.536T | 33.521s | 11.298 |
| Inference Method | Caching Modules | MACs | Latency | Generation Performance (CLIP) |
| OpenSora (250 inference steps) | - | 8417.500T | 178.913m | **31.211** |
| OpenSora + TGATE ($m$=100) | CA | 5696.500T | 120.826m | 30.779 |
| OpenSora + TGATE ($m$=100, $k$=3) | CA,SA | 5370.790T | 94.393m | 30.766 |
| OpenSora + TGATE ($m$=100, $k$=5) | CA,SA | 5298.410T | 88.519m | 30.683 |
| OpenSora + TGATE ($m$=100, $k$=10) | CA,SA | **5246.710T** | **84.323m** | 30.161 |

## 6.1 Improvement over Transformer-based Models

The proposed method is integrated into PixArt-Alpha (Chen et al., 2024b), a text conditional model based on transformer architecture (Peebles & Xie, 2023; Vaswani et al., 2017) [2]. As shown in Table 1, TGATE considerably accelerates the inference speed across various configurations. Notably, by setting $m$ to 15, TGATE enhances the efficiency and slightly reduces the FIDs to 9.548. Additionally, configuring TGATE with $m = 10$ and $k = 5$ further reduces computational demands while only moderately impacting the performance. This configuration only requires 64.138T MACs to generate a single image, thereby reducing the latency by nearly in half—from 61.502s to 32.827s. Notably, existing studies such as DeepCache (Ma et al., 2024) and BlockCache (Wimbauer et al., 2023) rely on the skipping architecture of U-Net (Ronneberger et al., 2015) to cache features for acceleration. However, these approaches are unsuitable for transformers due to their different architectural demands. To address this research gap, TGATE decomposes the contribution of attention mechanisms and reuses the features in redundant steps. This represents the first step to freely accelerate transformers by caching features.

Additionally, TGATE and its base models are qualitatively compared herein. Fig. 6 shows the images generated by different base models with or without TGATE. Although TGATE increases FID scores in some configurations, changes in generated samples are nearly imperceptible and demonstrate the effectiveness of TGATE in maintaining performance. More visualizations are available in Appendix E, and more analysis, including frame consistency, text-image alignment and memory cost analysis, is provided in Appendices H, I and J.

Table 2: Computational complexity, latency, and FID on the MS-COCO validation set using the base model of SD-1.5, SD-2.1, and SDXL. MACs stands for Multiply–Accumulate Operations per image. These terms are automatically generated using Calflops. The latency of generating one image is tested on a 1080 Ti commercial card. FID (Heusel et al., 2017) is used to test the performance of text-to-image models (Rombach et al., 2022; Podell et al., 2023) and CLIP score (Radford et al., 2021) for video models (Blattmann et al., 2023).

| Inference Method | MACs | Latency | Generation Performance (FID) |
|---|---|---|---|
| SD-1.5 | 16.938T | 7.032s | 23.927 |
| SD-1.5 + **TGATE** ($m$=5) | **9.875T** | **4.313s** | **20.789** |
| SD-1.5 + **TGATE** ($m$=10) | 11.641T | 4.993s | 23.269 |
| SD-2.1 | 38.041T | 16.121s | 22.609 |
| SD-2.1 + **TGATE** ($m$=5) | **22.208T** | **9.878s** | **19.940** |
| SD-2.1 + **TGATE** ($m$=10) | 26.166T | 11.372s | 21.294 |
| SDXL | 149.438T | 53.187s | 24.628 |
| SDXL + **TGATE** ($m$=5) | 84.438T | 27.932s | 22.738 |
| SDXL + **TGATE** ($m$=10) | 100.688T | 34.246s | 23.433 |
| SDXL + **TGATE** ($m$=5 $k$=3) | **83.498T** | **27.412s** | **22.306** |
| SDXL + **TGATE** ($m$=10 $k$=3) | 96.928T | 32.164s | 22.763 |
| SDXL + **TGATE** ($m$=10 $k$=5) | 95.988T | 31.643s | 23.839 |
| Inference Method | MACs | Latency | Generation Performance (CLIP) |
| SVD | 1609.250T | 645.842s | 31.322 |
| SVD + **TGATE** ($m$=5) | 935.650T | 408.485s | 31.176 |
| SVD + **TGATE** ($m$=10) | 1104.050T | 467.824s | 31.334 |
| SVD + **TGATE** ($m$=5 $k$=3) | **932.780T** | **402.044s** | 31.167 |
| SVD + **TGATE** ($m$=10 $k$=3) | 1092.570T | 442.060s | **31.358** |
| SVD + **TGATE** ($m$=10 $k$=5) | 1089.700T | 435.619s | 31.343 |

## 6.2 Improvement over U-Net-based Models

TGATE can also be applied to U-Net-based models (Rombach et al., 2022; Podell et al., 2023). For all settings shown in Table 2, TGATE enhances the performance of the base models in terms of computational efficiency and FID scores. In particular, TGATE works better when the parameter size of the base model increases. In SDXL, TGATE can reduce

---

[2]A technology with rich history, which can be dated back to the principles of the 1991 unnormalized linear Transformer (Schmidhuber, 1992a; Schlag et al., 2021).

the latency by half on the commercial GPU card (from 53.187 to 27.412 s). This indicates the effectiveness and scalability of TGATE on U-Net-based diffusion models. Qualitative analysis is given in Fig. 6 and Appendix E, and the evaluation of text alignment is discussed in Appendix H.

## 6.3 Improvement over Acceleration Models

Table 3: Computational complexity, latency, and FID using the LCM distilled from SDXL Podell et al. (2023) and PixelArt-Alpha Chen et al. (2024b).

| Inference Method | MACs | Latency | FID $\downarrow$ |
|---|---|---|---|
| LCM (SDXL, $n$=4) | 11.955T | 3.805s | 25.044 |
| LCM (SDXL, $n$=6) | 17.933T | 5.708s | 25.630 |
| LCM (SDXL, $n$=8) | 23.910T | 7.611s | 27.413 |
| + TGATE ($m$=1) | **11.171T** | **3.533s** | 25.028 |
| + TGATE ($m$=2) | 11.433T | 3.624s | **24.595** |
| LCM(PixArt, $n$=4) | 8.563T | 4.733s | **36.086** |
| + TGATE ($m$=1) | **7.623T** | **4.448s** | 38.574 |
| + TGATE ($m$=2) | 7.936T | 4.543s | 37.048 |

Table 4: Comparison with Adaptive Guidance (Castillo et al., 2023) on the MS-COCO validation set based on SDXL and Pixart-Alpha.

| Inference Method | MACs | Latency | FID $\downarrow$ |
|---|---|---|---|
| SDXL | 149.438T | 53.187s | 24.628 |
| w/ Adaptive Guidance | 104.606T | 35.538s | 23.301 |
| w/ TGATE ($m$=5, $k$=3) | **83.498T** | **27.412s** | **22.306** |
| PixelAlpha | 107.031T | 61.502s | 38.669 |
| w/ Adaptive Guidance | 74.922T | 42.684s | **35.286** |
| w/ TGATE ($m$=8) | 65.318T | 37.867s | 35.825 |
| w/ TGATE ($m$=10) | 70.225T | 40.648s | 35.726 |
| w/ TGATE ($m$=10,$k$ = 5) | **64.138T** | **32.827s** | 38.415 |

**Improvement over Consistency Model.** TGATE is implemented using a distillation-based method (Schmidhuber, 1992b; Hinton et al., 2015), namely the latent consistency model (LCM). The LCM distilled from SDXL [3] is first used as the base model, and a grid search is performed for different inference steps. Table 3 reveals that the generation performance is improved in fewer inference steps (i.e., four). To incorporate TGATE into the LCM, the cross-attention prediction of the first or second step ($m = 1, 2$) is cached and reused in the remaining inference steps. Due to limited inference steps, self-attention is not cached. Table 3 shows the experimental results of the LCM models distilled from SDXL and PixArt-Alpha[4]. Although the trajectory is deeply compressed into a few steps, TGATE functions well, and further decreases PixArt-based LCM computation. Thus, the MACs and latency are reduced by 10.98% and 6.02%, respectively, with comparable generation results. As TGATE does not incur any training costs, integrating it with consistency models is valuable and promising. As a reasonable blueprint, distillation-based methods require sampling trajectories from the teacher model; contrarily, TGATE can enhance sampling efficiency, thereby accelerating the learning process. This aspect will be further discussed in a future study. The visualization is provided in Appendix E.

**Comparison with Adaptive Guidance.** Adaptive guidance (Castillo et al., 2023) offers a strategy for the early termination of CFG. The efficiency of TGATE surpasses that of adaptive guidance, as it innovatively caches and reuses attention. On terminating the CFG in PixArt-Alpha, adaptive guidance yields 2.14T MACs/step, whereas TGATE skips cross-attention and further reduces this value to 1.83T MACs/step. This optimization moderately reduces the computational overhead, as shown in Table 4. Notably, recent trends have shifted toward distillation-based techniques (Song et al., 2023; Salimans & Ho, 2022) to accelerate inference. These methods compress the denoising process into fewer steps, often achieving single-digit iterations. The student model learns to mimic the CFG-based output during distillation; therefore, CFG decreases during inference, rendering adaptive guidance inapplicable. In contrast, TGATE can fill this gap and further accelerate the distillation-based models, as shown in Table 3. Beyond current capabilities, the superior scalability of TGATE is compared with that of adaptive guidance, particularly with increasing input sizes. This scalability feature of TGATE is further explored in Appendix D.

**Improvement over DeepCache.** Table 5 compares the performance of DeepCache (Ma et al., 2024) and TGATE based on SDXL. Although DeepCache is more efficient, TGATE outperforms it in terms of generation quality. TGATE is integrated with DeepCache by reusing cross-attention maps, thereby yielding superior results: MACs and latency of 43.868 T and 14.666 s. Remarkably, DeepCache caches the mid-level blocks to decrease computational load, which is specific to the U-Net architecture. However, its generalizability to other architectures, such as the transformer-based architecture, remains underexplored. Beyond DeepCache, TGATE has wider applications and can considerably improve transformer-based diffusion models, as shown in Table 1. Owing to this adaptability, TGATE is a versatile

---

[3]https://huggingface.co/latent-consistency/lcm-sdxl
[4]https://huggingface.co/PixArt-alpha/PixArt-LCM-XL-2-1024-MS

Table 5: Computational complexity, latency, and FID on the MS-COCO validation set using DeepCache (Ma et al., 2024).

| Inference Method | MACs | Latency | FID ↓ |
|---|---|---|---|
| SDXL | 149.438T | 53.187s | 24.628 |
| TGATE | 83.498T | 27.412s | **22.306** |
| DeepCache | 57.888T | 19.931s | 23.755 |
| DeepCache + TGATE | **43.868T** | **14.666s** | 23.999 |

Table 6: Zero-shot FIDs on the MS-COCO validation set using the base model of SDXL and different noise schedulers (Karras et al., 2022; Lu et al., 2022; Song et al., 2021).

| Scheduler | Base Model | TGATE(CA) | TGATE(CA,SA) |
|---|---|---|---|
| EulerD | 23.084 | 21.883 | **21.857** |
| DDIM | **21.377** | 21.783 | 21.821 |
| DPMSolver | 24.628 | 22.738 | **22.306** |

and potent enhancement for various architectural frameworks. Self-attention is fully operational in the experiment conducted herein and aligned with the DeepCache strategy, as it is not included in the first and last blocks.

**Improvement over Different Schedulers.** The generalizability of TGATE is evaluated on different noise schedulers. As shown in Table 6, three advanced schedulers (Karras et al., 2022; Lu et al., 2022; Song et al., 2021), are considered that could compress the generation process of a diffusion model to 25 inference steps. Results show that TGATE could consistently achieve stable generation performance in all settings, further indicating its potential for a broad application.

**Additional Comparison with other methods.** Beyond the previously discussed methods, we explore integrating our approach with other accelerated diffusion models: (1) SSD (Gupta et al., 2024), which reduces model size via distillation from a larger model, and (2) ToMe (Bolya & Hoffman, 2023), which accelerates inference by compressing token counts. Further details are provided in Appendix G.

# 7 Conclusion and Discussion

The cross-attention and self-attention in the inference process of text-conditional diffusion models are empirically analyzed here, offering the following: i) In the first few inference steps, cross-attention is the primary contributor; however, its influence diminishes in later steps. ii) In contrast, self-attention plays a secondary role initially but gains significance as denoising progresses. iii) By caching and reusing attention maps during scheduled inference steps, TGATE reduces computational demands while still achieving competitive outcomes. These findings encourage further analysis of the role of attention in text-conditional diffusion models.

**Limitations** We acknowledge the challenge of further improving a distilled diffusion model when working with few inference steps. In such cases, TGATE can only offer a 10% reduction in computation cost (MACs) for models with highly compressed inference steps, such as LCM. However, considering that a model may be used billions of times daily, this reduction is significant. The proposed TGATE is particularly valuable since it does not require additional training costs or result in significant performance drops. Furthermore, TGATE can also be integrated into the distillation process, where it accelerates the generation process of the teacher model, potentially speeding up the training of the student model.

Empirical studies suggest that using TGATE may cause a slight decline in text-image alignment performance (Sec. H) but generally improves the FID score (Sec. 6.1). Based on visualizations, we speculate that although TGATE-generated images are similar to those without it, they tend to produce simpler patterns and objects, akin to outcomes seen in other acceleration methods. To address this, we provide a set of parameters ($k$ and $m$) to balance efficiency and performance, allowing adaptation to different application scenarios. Despite these trade-offs, TGATE reduces inference time by nearly half, making it an effective solution for most cases.

**Broader Impacts**

**Positive Broader Impacts** The text-conditional diffusion model, which may be used billions of times daily, typically requires extensive energy resources. Without incurring additional training costs, the computational demands of various base models can be reduced by 10% – 50% using the proposed approach. Thus, it is an eco-friendly solution that considerably reduces the electricity consumption associated with AI technologies.

**Negative Broader Impacts** This study is fundamental and not linked to specific applications. Therefore, the negative social impacts associated with TGATE are consistent with those of other text-conditional diffusion models and do not present unique risks that warrant a specific mention here.

## Acknowledgment

We thank Dylan R. Ashley, Bing Li, Haoqian Wu, Yuhui Wang, and Mingchen Zhuge for their valuable suggestions, discussions, and proofreading.

Jinheng Xie and Mike Shou are only supported by the Ministry of Education, Singapore, under its Academic Research Fund Tier 2 (Award No: MOE-T2EP20124-0012).

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

# A  Additional Experiments for Temporal Analysis of Cross-Attention

Table A1: Zero-shot FIDs on the MS-COCO validation set using the base model SD-2.1 with DPM-Solver (Lu et al., 2022). $m$ is the gate step.

| Configuration | FID ↓ |
|---|---|
| SD-2.1(w/ CFG) | 22.609 |
| $\mathbf{S}_m^F$ ($m = 3$) | 29.282 |
| $\mathbf{S}_m^F$ ($m = 5$) | **20.859** |
| $\mathbf{S}_m^F$ ($m = 10$) | 21.816 |

Table A2: Zero-shot FIDs on the MS-COCO validation set using the base model SD-2.1 with DPM-Solver (Lu et al., 2022). $m$ is the gate step and $n$ is the total inference number.

| Configuration | S | $\mathbf{S}_m^F$ | $\mathbf{S}_m^L$ |
|---|---|---|---|
| $n$=15, $m$=6 | 23.380 | 22.315 | 58.580 |
| $n$=25, $m$=10 | 22.609 | 21.816 | 53.265 |
| $n$=50, $m$=20 | 22.445 | 21.557 | 48.376 |
| $n$=100, $m$=25 | 22.195 | 20.391 | 26.413 |

Table A3: Zero-shot FIDs on the MS-COCO validation set using the base model SD-2.1 with different noise schedulers. The total inference number is set as 50, and the gate step is 20.

| Configuration | S | $\mathbf{S}_m^F$ | $\mathbf{S}_m^L$ |
|---|---|---|---|
| EulerD (Karras et al., 2022) | 22.507 | 21.559 | 47.108 |
| DPMSolver (Lu et al., 2022) | 22.445 | 21.557 | 48.376 |
| DDIM (Song et al., 2021) | 21.235 | 20.495 | 53.150 |

Table A4: Zero-shot FIDs on the MS-COCO validation set using different models: SD-1.5, SD-2,1, and SDXL. The total inference number is set as 25, and the gate step is 10.

| Configuration | S | $\mathbf{S}_m^F$ | $\mathbf{S}_m^L$ |
|---|---|---|---|
| SD-1.5 (Rombach et al., 2022) | 23.927 | 22.974 | 37.271 |
| SD-2.1(Rombach et al., 2022) | 22.609 | 21.816 | 53.265 |
| SDXL (Podell et al., 2023) | 24.628 | 23.195 | 108.676 |

Additional experiments are performed for different gate steps of {3,5,10} to support the analysis on cross-attention. As shown in Table A1, when the gate step is larger than five steps, the model that ignores cross-attention can achieve better FIDs. To further justify the generalization of these findings, experiments are conducted under various conditions, including a range of total inference numbers, noise schedulers, and base models. Table A2, A3, and A4 show that FIDs of $\mathbf{S}$, $\mathbf{S}_m^F$, and $\mathbf{S}_m^L$ on the MS-COCO validation set.

# B  Temporal Analysis of Self-Attention

Table A5: Zero-shot FIDs on MJHQ-10k using the base model PixArt-Alpha with different caching and re-using strategies.

| Trajectory | $k$ | $m$ | Total Inference Steps | FIDs |
|---|---|---|---|---|
| **S** | 1 | - | 25 | 9.653 |
| Inference steps in the semantics-planning phase is less than that in the fidelity-improving phase. | | | | |
| **S** | 1 | 10 | 25 | 11.268 |
| $\mathbb{S}^F$ | 3 | 10 | 25 | 19.205 |
| $\mathbb{S}^L$ | 3 | 10 | 25 | **11.789** |
| $\mathbb{S}^F$ | 5 | 10 | 25 | 29.507 |
| $\mathbb{S}^L$ | 5 | 10 | 25 | **12.738** |
| Inference steps in the semantics-planning phase is larger than that in the fidelity-improving phase. | | | | |
| **S** | 1 | 15 | 25 | 9.548 |
| $\mathbb{S}^F$ | 3 | 15 | 25 | 11.436 |
| $\mathbb{S}^L$ | 3 | 15 | 25 | **10.289** |
| Inference steps in the semantics-planning phase is equal to that in the fidelity-improving phase. | | | | |
| **S** | 1 | 10 | 20 | 10.105 |
| $\mathbb{S}^F$ | 3 | 10 | 20 | 17.000 |
| $\mathbb{S}^L$ | 3 | 10 | 20 | **11.482** |

Here, the functionality of self-attention in the denoising trajectories from a pre-trained diffusion model is explored.

For a comprehensive study, we track the generation performance using different values of gate step $m$ and interval $k$. As shown in Table A5, the empirical results convincingly show that self-attention plays a vital role in the latter phases of the process. This is evidenced by consistently higher FID score for $\mathbb{S}^F$ than that for $\mathbb{S}^L$ across all tests.

## C   Implementation Details

**Base Models.**  Several pre-trained models are used in the experiments: Stable Diffusion-1.5 (SD-1.5) (Rombach et al., 2022), SD-2.1 (Rombach et al., 2022), SDXL (Podell et al., 2023), PixArt-Alpha (Chen et al., 2024b), SVD (Blattmann et al., 2023) and OpenSora (Lab & etc., 2024). Among them, the SD series are based on convolutional neural networks (Fukushima, 1979; 1980; Zhang et al., 1988; LeCun et al., 1989; Hochreiter, 1991; Srivastava et al., 2015; He et al., 2016) (*i.e.*, U-Net (Ronneberger et al., 2015)). Pixart-Alpha and OpenSora work on the transformer (Vaswani et al., 2017) (*i.e.*, DiT(Peebles & Xie, 2023)). This experimental setting covers several conditional generation tasks, including text-to-image, text-to-video, and image-to-video tasks.

**Acceleration Baselines.**  For a convincing empirical study, TGATE is compared with several acceleration baseline methods: Latent Consistency Model (Luo et al., 2023), Adaptive Guidance (Castillo et al., 2023), DeepCache (Ma et al., 2024), and multiple noise schedulers (Karras et al., 2022; Lu et al., 2022; Song et al., 2021). TGATE is orthogonal to existing methods used to accelerate denoising inference; therefore, it can be trivially integrated to further accelerate this process.

**Evaluation Metrics.**  Similar to a previous study (Podell et al., 2023), 10k images from the MS-COCO validation set (Lin et al., 2014) are used to evaluate the zero-shot generation performance. The images are generated using DPM-Solver (Lu et al., 2022) with a predefined 25 inference steps and resized to $256 \times 256$ resolution to calculate the FID (Heusel et al., 2017). TGATE is also tested on a high-resolution dataset, namely MJHQ (Li et al., 2023) to evaluate its aesthetic quality. The generated images are set at a resolution of $1024 \times 1024$, with a total of 10k samples. Following the protocol in ELLA (Hu et al., 2024), we utilize mPLUG-Large (Li et al., 2022) to score the generated samples based on predefined questions. For video generation, the prompts from OpenSora-Sample (Lab & etc., 2024) are used. 10 videos per prompt are generated, and their performance is evaluated based on the CLIP score (Radford et al., 2021). As the text branch of SVD is unavailable, SDXL is used to create an image from a prompt, which is then input into SVD to produce the corresponding video. To evaluate the efficiency, Calflops (xiaoju ye, 2023) is used to count Multiple-Accumulate Operations (MACs) and the number of parameters (Params.). Furthermore, the latency per sample is assessed on a platform equipped with a Nvidia 1080 Ti.

## D   Discussion on Scaling Token Length and Resolution

Table A6: MACs per inference step when scaling up the token lengths and image resolutions.

| Resolution | Method | Tokens Scaling Factor | | | |
|---|---|---|---|---|---|
| | | $\times 1$ | $\times 128$ | $\times 1024$ | $\times 4096$ |
| 768 | SD-2.1 (w/o CFG) | 0.761T | 1.011T | 2.774T | 8.820T |
| | Ours | **0.73**T | | | |
| 1024 | SD-2.1 (w/o CFG) | 1.351T | 1.601T | 3.364T | 9.410T |
| | Ours | **1.298**T | | | |
| 2048 | SD-2.1 (w/o CFG) | 5.398T | 5.648T | 7.411T | 13.457T |
| | Ours | **5.191**T | | | |

TGATE can improve efficiency by circumventing cross-attention, which motivated us to examine its contribution to the overall computational cost based on the input size. Specifically, TGATE is compared with SD-2.1 w/o CFG per step to determine the computational cost of cross-attention. Note that SD-2.1 w/o CFG is the computational lower bound for existing methods (Castillo et al., 2023), as it can stop CFG early to accelerate diffusion process. As shown in Table A6, MACs are used as a measure of efficiency. In the default SD-2.1 setting, the resolution is set as 768 with a maximum token length of 77. Results show that cross-attention moderately contributes to the total computational load, which increases exponentially with increasing resolution and token lengths. By omitting cross-attention calculations, TGATE

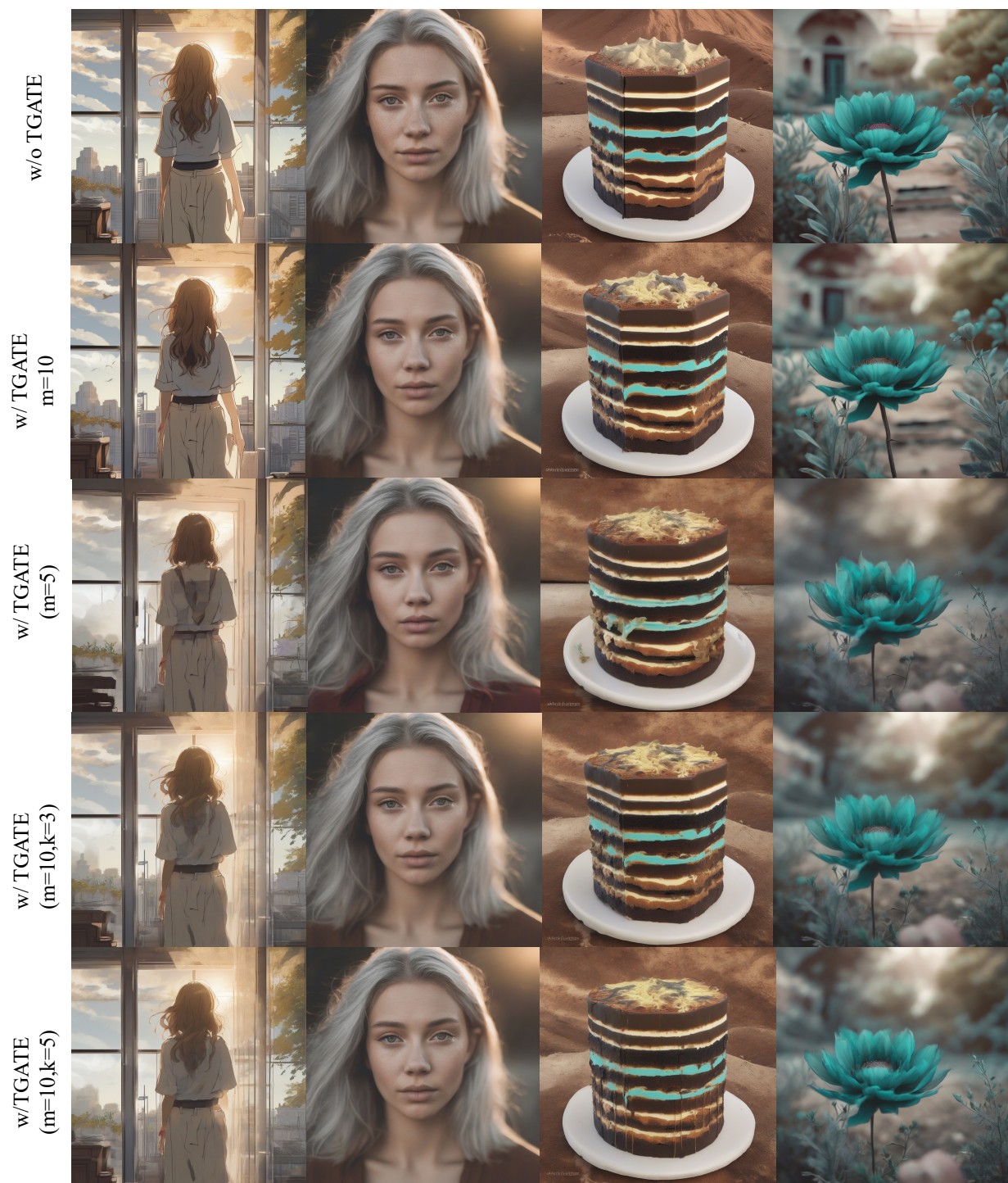

Figure A1: **Generated samples by SDXL using the same initial noises and prompts, but with varying hyper-parameters.** The configurations are arranged from top to bottom in the order of decreasing latency.

PixArt-$\alpha$

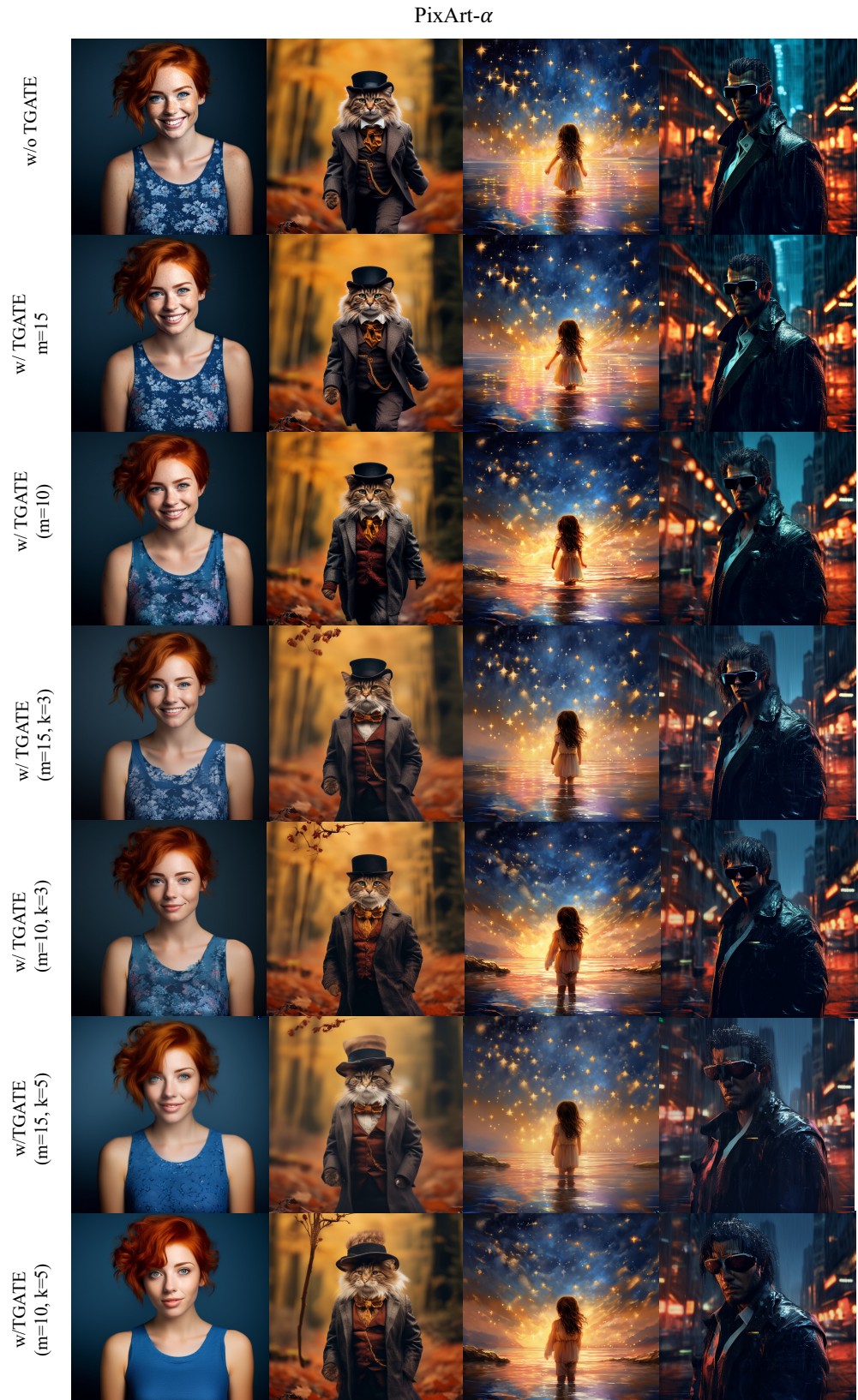

Figure A2: **Samples generated by PixArt using the same initial noises and prompts and different hyperparameters.** The configurations, from top to bottom, are ordered by decreasing latency.

considerably mitigates its adverse effects. For example, in an extreme scenario with the current architecture targeting an image size of 2048 and a token length of 4096 × 77, MACs can be decreased from 13.457 T to 5.191 T, achieving more than two-fold reduction in computation.

One may argue that existing models do not support such high resolutions or token lengths. However, there is an inevitable trend toward larger input sizes (Zhang et al., 2024; Esser et al., 2024; Chen et al., 2024a). Furthermore, a recent study (Li et al., 2024) has shown the difficulty in computing cross-attention on mobile devices, underscoring the practical benefits of our approach.

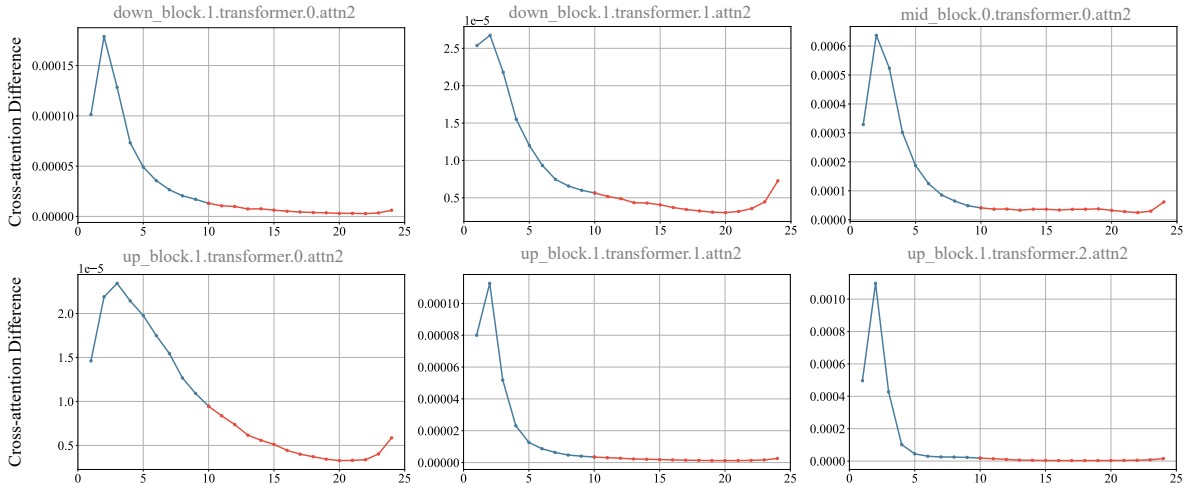

Figure A3: Illustration of cross-attention map differences between consecutive inference steps. We sample cross-attention maps from various blocks in SD-2.1 to monitor their convergence. The consistent convergence across different blocks supports the generality of our observations.

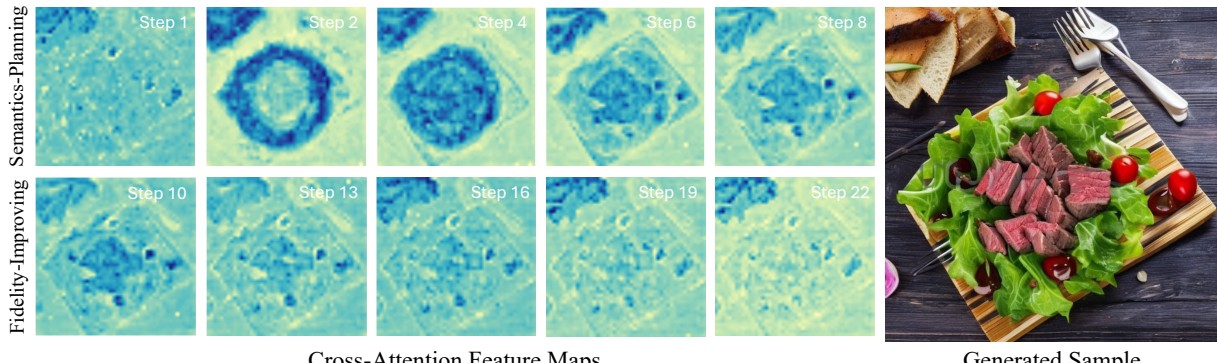

Figure A4: Illustration of cross-attention maps during different inference steps. We use up_block.1.transformer_block.2.attn2 in SD-2.1 as a representative module to visualize its output for a given prompt, "a delicious salad with beef, stock photo". During inference, the feature map converges rapidly and stabilizes, which aligns with our observations.

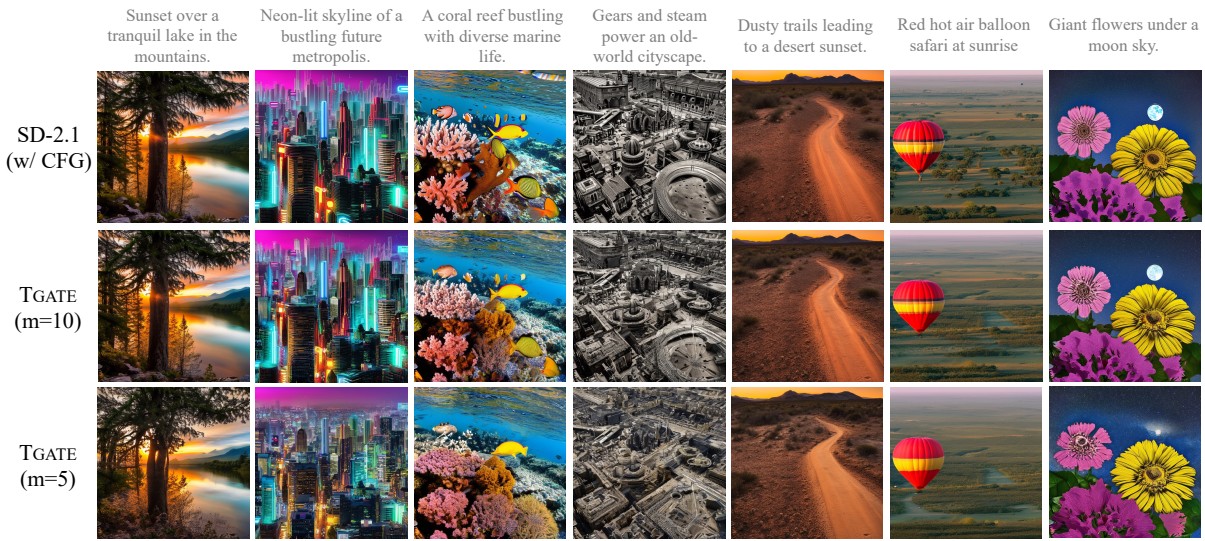

Figure A5: Samples generated by SD-2.1 (w/ CFG) and TGATE with two gate steps (*i.e.*, 5 and 10) for the same initial noises and captions.

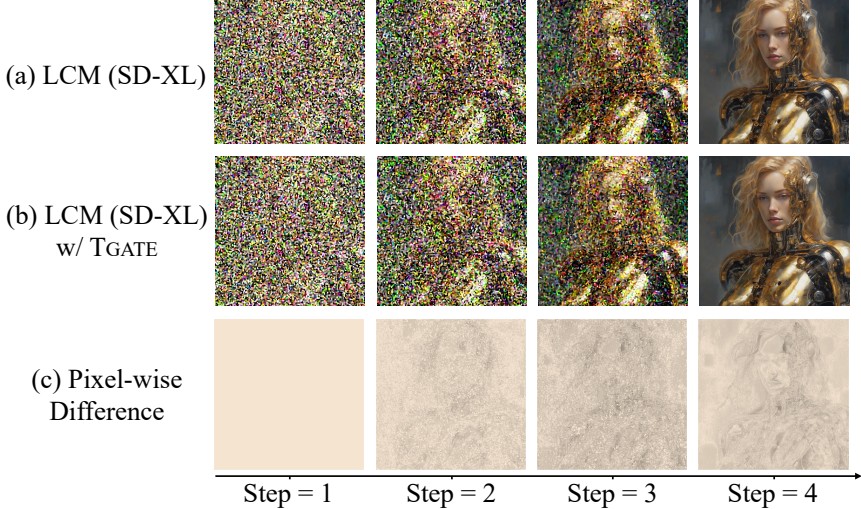

Figure A6: Generated samples of (a) LCM distilled from SDXL and (b) LCM with TGATE given the same initial noise and captions. (c) represents the difference between (a) and (b).

# E    Additional Visualization

To support Sec. 3, we further analyze the distribution of different blocks' attention modules. As shown in Fig. A3, we sample the attention maps from `down_blocks`, `mid_block`, and `up_blocks`, and track their changes during different steps. The convergence rates are slightly different, but they share the same trend, which further indicates that this interesting phenomenon is common among different blocks. Meanwhile, as shown in Fig. A4, we also visualize the feature map of `up_block` as the representative one during different inference steps.

We also provide visualizations of TGATE with different models and configurations. Fig. A5 shows the samples generated using different gate steps. Results show that larger gate steps produce generation results more similar to those of the base models without TGATE. Moreover, the generated samples are visualized based on different steps. As shown in Fig. A6, the difference caused by TGATE is invisible. Considering that different configurations have moderate impacts on FIDs, the samples generated by SDXL and PixArt-Alpha are visualized under various settings, as shown

in Fig. A1 and Fig. A2. Although some configurations result in increased FIDs, the changes are nearly imperceptible. These results demonstrate the effectiveness of TGATE.

Table A7: Generation performance on MJHQ-10k (Li et al., 2023) and MS-COCO (Lin et al., 2014) datasets with different random seeds on PixArt (Chen et al., 2024b) and SDXL (Podell et al., 2023).

|  | Dataset | Caching Modules | Seed 1 | Seed 2 | Seed 3 | Mean ± S.d. |
|---|---|---|---|---|---|---|
| PixArt | MJHQ-10K | - | 9.653 | 9.642 | 9.539 | 9.611 ± 0.051 |
| TGATE | MJHQ-10K | CA | 9.548 | 9.477 | 9.417 | **9.481 ± 0.054** |
| TGATE | MJHQ-10K | CA, SA | 10.289 | 10.108 | 10.138 | 10.178 ± 0.079 |
| SDXL | COCO-10K | - | 24.628 | 24.164 | 24.661 | 24.484 ± 0.227 |
| TGATE | COCO-10K | CA | 23.433 | 22.917 | 23.584 | **23.311 ± 0.286** |
| TGATE | COCO-10K | CA, SA | 23.839 | 22.759 | 23.967 | 23.522 ± 0.542 |

## F  Error Bar

We provide error bars for the main experiments with configurations of $m = 15$, $k = 3$ for PixArt and $m = 10$, $k = 5$ for SDXL. Results in Table 1 and Table 2 are based on random seed 1.

## G  Additional Comparison

We further incorporate our method into SSD-1B Gupta et al. (2024), a lightweight model distilled from a larger diffusion model. As shown in Table C, we test the models' performance on the COCO Validation Set in terms of latency and FID-10k. Latency is defined as the time required to generate one image in a resolution of 768, and the results are collected on a platform using an Nvidia 1080ti. The results demonstrate that our method is also compatible with models distilled from larger models.

We also compare our method with ToMe Bolya & Hoffman (2023). We set ToMe's merging ratio to 50% and use SD-2.1 as the base model. The model utilizes the DPM-Solver

Table A8: Generation performance of ToMe (Bolya & Hoffman, 2023) and SSD-1B (Gupta et al., 2024) with and without with TGATE on MS-COCO-10k (Lin et al., 2014) dataset. The base mode for ToMe is SD-2.1(Rombach et al., 2022).

|  | FID | Latency |
|---|---|---|
| ToMe | 24.763 | 12.240s |
| TGATE + ToMe | **24.383** | **8.731s** |
| SSD-1B | 28.564 | 30.097s |
| TGATE | **25.004** | **18.328s** |

with 25 inference steps as the noise scheduler. Our method achieves a latency of 11.372 seconds per image, slightly outperforming ToMe. Given that our approach is orthogonal to ToMe, integrating TGATE with ToMe results in an improved latency of 8.731 seconds. This empirical study demonstrates the effectiveness of our method and its potential for broad application when combined with various acceleration techniques.

## H  Evaluation on Text-Image Alignment

The performance of TGATE in text-image alignment is assessed based on the protocol outlined by CLIP score Radford et al. (2021), ELLA (Hu et al., 2024) and T2I-Compbench (Huang et al., 2023). Table A9 and Table A10 shows the competitive performance of TGATE compared with the baseline across various evaluation dimensions, indicating its effectiveness. We provide visualization samples in Fig. A7. As noted in our limitations, increased inference acceleration can lead to greater deviations from the baseline. While these differences are often perceptually indistinguishable, in some cases, they may cause noticeable artifacts, such as the altered shape of the bowl or slight distortion of the bracelet in Fig. A7. Given the acceleration achieved, this trade-off is acceptable in most cases that do not demand exceptionally high-quality samples.

Table A9: Generation performance on DPG-Bench (Hu et al., 2024). CA and SA represent cross-attention and self-attention, respectively. For PixArt, parameters are set to $m = 15$ and $k = 3$, whereas for SDXL, we utilize $m = 10$ and $k = 5$. Evaluation scores are obtained using mPLUG-Large (Li et al., 2022) with predefined questions.

| Method | Global↑ | Entity↑ | Attribute↑ | Relation↑ | Other↑ | Total Score↑ |
|---|---|---|---|---|---|---|
| PixArt | 78.81 | **79.46** | 80.61 | 78.22 | **80.24** | **72.53** |
| TGATE (CA) | **79.31** | 78.44 | **79.33** | **80.68** | 75.48 | 70.55 |
| TGATE (CA, SA) | 76.67 | 76.18 | 77.56 | 78.74 | 79.14 | 68.37 |
| SDXL | **81.87** | **82.50** | **82.46** | 83.50 | 61.45 | **74.92** |
| TGATE (CA) | 70.33 | 80.32 | 80.60 | **83.63** | 78.63 | 72.41 |
| TGATE (CA, SA) | 79.65 | 77.29 | 78.43 | 82.78 | **80.11** | 70.72 |

Table A10: Generation performance of SD-2.1 (Rombach et al., 2022) with TGATE on T2I-Compbench (Huang et al., 2023). We set $m$ as 10 and $k$ as 5.

| | Color | Shape | Texture | Spatial | Non-Spatial | Complex |
|---|---|---|---|---|---|---|
| SD-2.1 | 43.31 | 42.27 | 47.24 | 6.52 | 29.80 | 30.50 |
| SD-2.1 + TGATE (CA) | 41.97 | 41.16 | 45.28 | 5.73 | 29.55 | 29.45 |
| SD-2.1 + TGATE (CA,SA) | 40.72 | 40.01 | 44.22 | 5.14 | 29.32 | 29.44 |

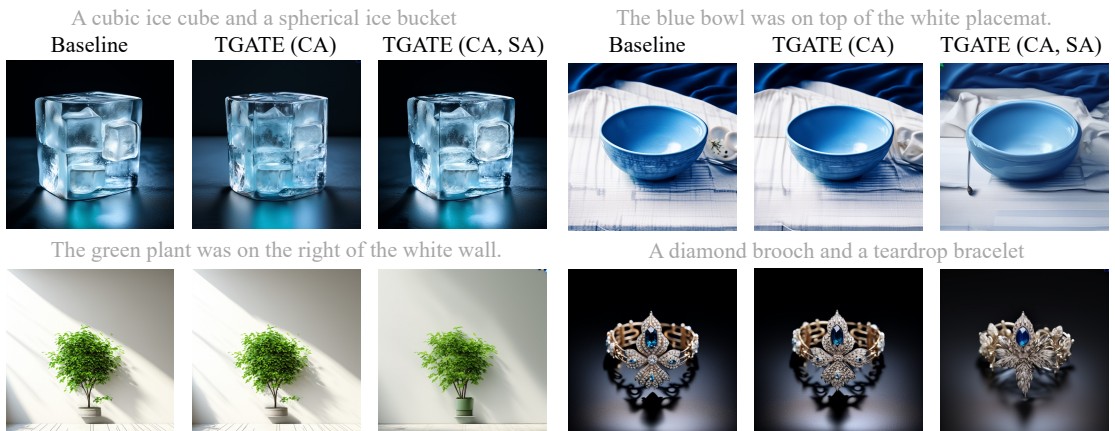

Figure A7: Generated samples from (a) PixArt, (b) TGATE with CA cache, and (c) TGATE with both CA and SA cache. The caption is derived from T2I-Coompbench.

Table A11: CLIP score of TGATE on MJHQ (Li et al., 2023) and MS-COCO (Lin et al., 2014) datasets on PixArt (Chen et al., 2024b) and SDXL (Podell et al., 2023).

| Method | CLIP Score |
|---|---|
| SDXL | 34.148 |
| TGATE ($m$=10) | 33.371 |
| TGATE ($m$=10 $k$=5) | 32.710 |
| PixArt-Alpha | 34.233 |
| TGATE ($m$=15) | 33.872 |
| TGATE ($m$=15 $k$=3) | 33.673 |

Table A12: Frame consistency on Open-Sora sample dataset (Lab & etc., 2024). The frame consistency is calculated based on the L2 distance between the adjacent frames.

| Model | Frame Consistency |
|---|---|
| OpenSora | 9.979 |
| TGATE ($m = 100$) | **7.597** |
| TGATE ($m = 100, k = 3$) | 9.987 |
| SVD | **30.686** |
| TGATE ($m = 10$) | 31.162 |
| TGATE ($m = 10, k = 5$) | 31.020 |

Table A13: Memory overhead caused by TGATE. The base model is PixArt (Chen et al., 2024b) and SDXL (Podell et al., 2023) and the computational platform is a single V100 GPU card with pytorch 2.2.

| | Memory Cost |
|---|---|
| SDXL | 8515 MB |
| TGATE ($m$=10, $k$=5) | 8531 MB |
| PixArt-Alpha | 13705 MB |
| TGATE ($m$=15, $k$=3) | 13707 MB |

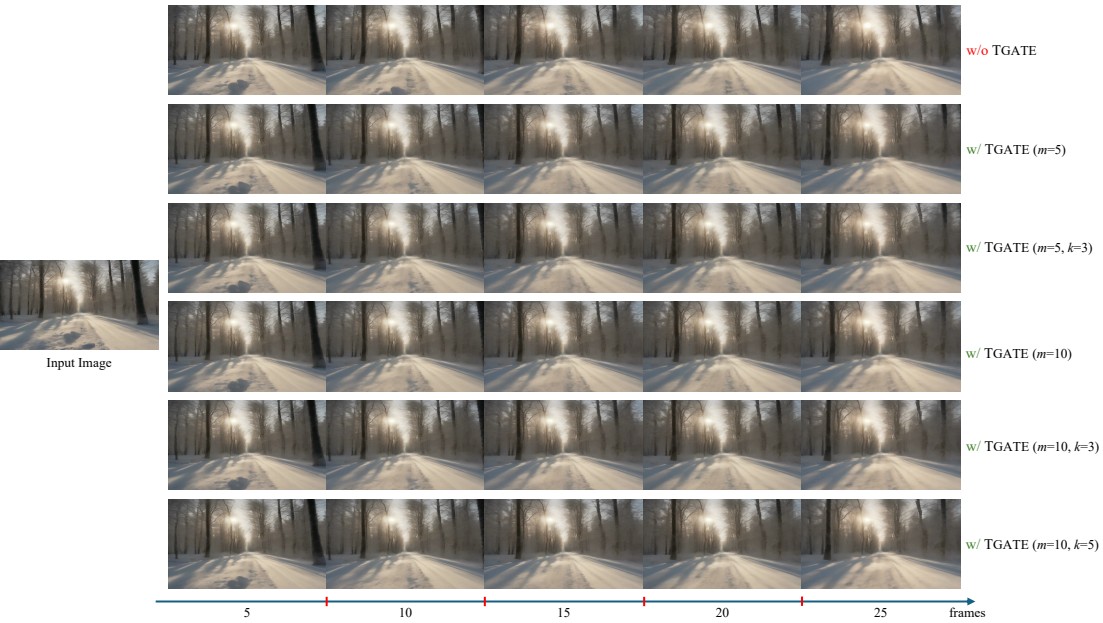

Figure A8: Generated samples based on SVD. SDXL generates the input frame with a caption of "A snowy forest landscape with a dirt road running through it. The road is flanked by trees covered in snow, and the ground is also covered in snow. The sun is shining, creating a bright and serene atmosphere. The road appears to be empty, and there are no people or animals visible in the video. The style of the video is a natural landscape shot, with a focus on the beauty of the snowy forest and the peacefulness of the road.".

## I    Evaluation on Frame Consistency

The performance of TGATE in video generation is evaluated based on the frame consistency. The L2 distance between different frames are obtained, where a smaller value indicates a smoother change between frames. As shown in Table A12, TGATE does not cause significant differences in this metric, confirming its effectiveness. We also provide a visualization in Fig. A8

## J    Evaluation on Memory Cost

We evaluate the memory overhead of TGATE, which accelerates inference by caching features and may increase GPU usage. As shown in Table A13, using SDXL and Pixart-Alpha as baselines for U-Net and transformer models, TGATE incurs only a minimal, single-digit memory cost, which is negligible in most cases, demonstrating its feasibility.

## K    Discussion on Anchor Feature.

This paper employs the average feature maps from both unconditional and conditional branches as the anchor feature for caching, following CFG. We provide visualization samples using only one branch instead of averaging. As shown in Fig. A9, the results suggest minimal impact on performance. This aligns with our pilot study (Fig. 2), where attention maps converge to a fixed point after several inference steps, including for the conditional/unconditional branches in CFG.

## L    Hyper-parameter Selection

TGATE is a simple method for training-free acceleration with two hyperparameters, $k$ and $m$. Their values can adapt to different inference steps to achieve a balanced trade-off. As shown in Fig. A10, $m$ and $k$ are set as $3/5$ and $1/5$ of the total inference steps, respectively. Increasing inference steps minimally impacts the generated results, suggesting that $k$ and $m$ can be reliably set as fixed proportions of the inference steps for most scenarios.

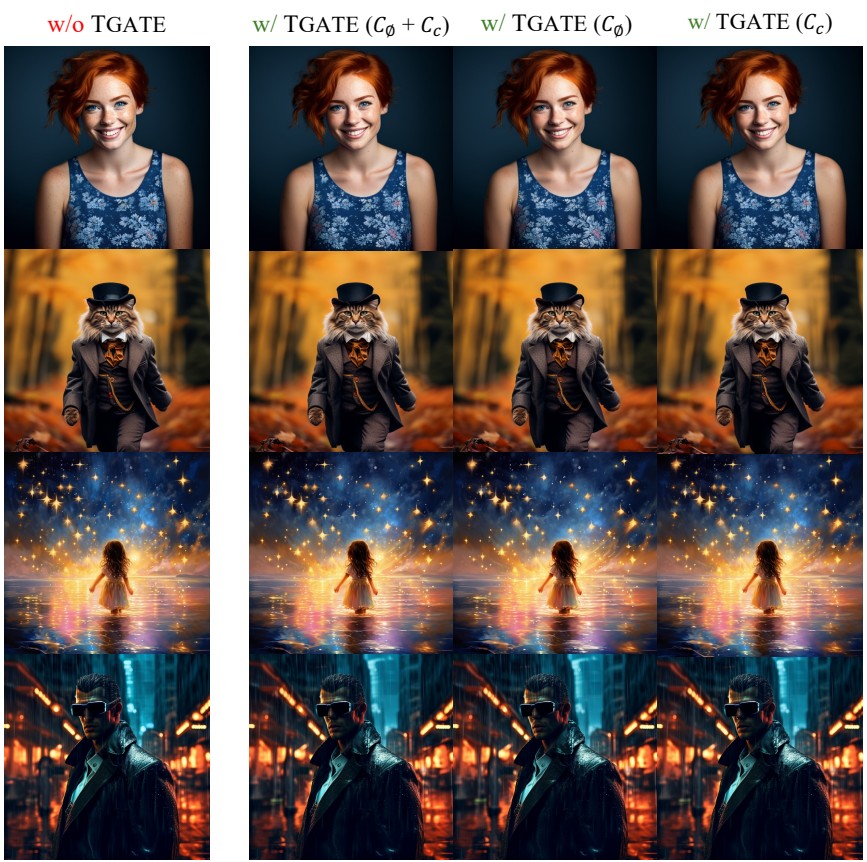

Figure A9: Generated samples of (a) PixArt and (b) TGATE reusing the averaged of cross-attention maps, (c) TGATE reusing the unconditional cross-attention maps, and (d) TGATE using text-conditional cross-attention maps.

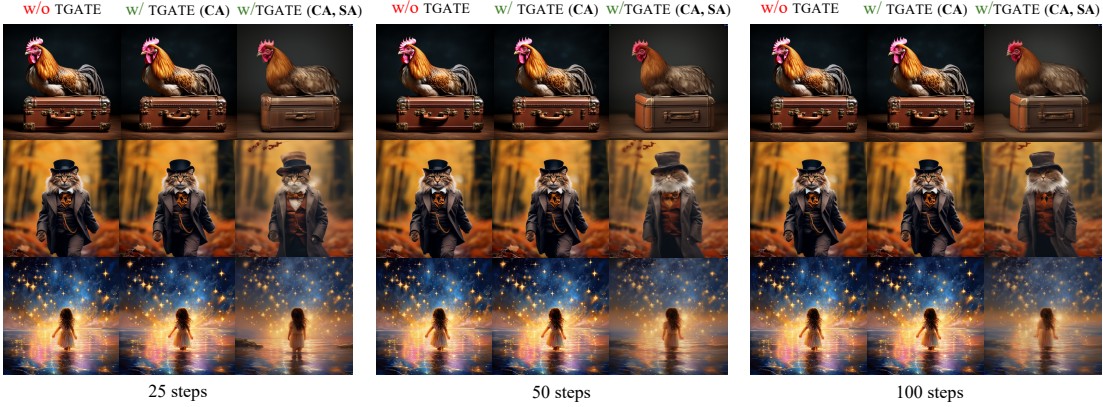

Figure A10: Generated samples of PixArt are evaluated with 25, 50, and 100 inference steps. TGATE's hyperparameter, set as ratios of inference steps ($m = \frac{3}{5}$inference-steps and $k = \frac{1}{5}$inference-steps) ensure stable performance across different settings.

## M   Details of Prompt

We showcase all prompts used in this paper to ensure reproducibility.

Table A14: Figures and Corresponding Prompts

| Figure | Caption |
|---|---|
| Figure 1 | 'A desert oasis with a mirage of a city.'
'A galaxy with colorful nebulae and star clusters.'
'An underwater coral reef with mermaids and treasure.'
'A magical duel in a ruined castle.'
'A painting of mountains, in the style of Monet.' |
| Figure 3 | 'High quality photo of an astronaut riding a horse in space.' |
| Figure 4 | 'Paisaje montañoso nevado.' |
| Figure 6 (a) | 'Beautiful lady, freckles, big smile, blue eyes, short ginger hair, dark makeup, wearing a floral blue vest top, soft light, dark grey background.'
'Professional portrait photo of an anthropomorphic cat wearing fancy gentleman hat and jacket walking in autumn forest.'
'Stars, water, brilliantly, gorgeous large scale scene, a little girl, in the style of dreamy realism, light gold and amber, blue and pink, brilliantly illuminated in the background.'
'A legendary figure in Cyberpunk City, rainy night, front, sunglasses.' |
| Figure 6 (b) | 'Woman standing in front of a window with her hair blowing, modern anime, sunshine, highly detailed.'
'Create a story or description inspired by this photo of a gray-haired 24 year old Norwegian woman in front of the sun with subtle freckles and hazel eyes.'
'A layer cake made out of a stratographic cross-section of the Sonoran Desert.'
'A teal flower in a barren garden, beautiful villa background, octane, redshift, highly detailed.' |
| Figure A4 | 'a delicious salad with beef, stock photo.' |
| Figure A5 | 'Sunset over a tranquil lake in the mountains'
'Neon-lit skyline of a bustling future metropolis. '
'A coral reef bustling with diverse marine life.'
'Gears and steam power an old-world cityscape.'
'Dusty trails leading to a desert sunset.'
'Red hot air balloon safari at sunrise'
'Giant flowers under a moon sky' |
| Figure A6 | 'Self-portrait oil painting, a beautiful cyborg with golden hair, 8k.' |
| Figure A8 | 'A snowy forest landscape with a dirt road running through it. The road is flanked by trees covered in snow, and the ground is also covered in snow. The sun is shining, creating a bright and serene atmosphere. The road appears to be empty, and there are no people or animals visible in the video. The style of the video is a natural landscape shot, with a focus on the beauty of the snowy forest and the peacefulness of the road.' |
| Figure A7 | 'A cubic ice cube and a spherical ice bucket.'
'The blue bowl was on top of the white placemat.'
'The green plant was on the right of the white wall.'
'A diamond brooch and a teardrop bracelet' |
| Figure A10 | 'a chicken on the right of a suitcase.'
"professional portrait photo of an anthropomorphic cat wearing fancy gentleman hat and jacket walking in autumn forest."
'stars, water, brilliantly, gorgeous large scale scene, a little girl, in the style of dreamy realism, light gold and amber, blue and pink, brilliantly illuminated in the background.' |

