# OpenReview forum: "Faster Diffusion Through Temporal Attention Decomposition"
_TMLR — Accepted by TMLR_

### Review · Reviewer_bq4G · 2024-11-01

**Summary Of Contributions:**

The paper presents experimental results demonstrating that diffusion sampling breaks into two phases, the first of which is dominantly influenced by cross-attention and the second of which is dominated by self-attention. The authors leverage this biphasic behavior by proposing a method to reduce computational demands by periodically caching and reusing self-attention maps during the first phase of sampling and caching cross-attention feature maps (which converge after the early part of sampling) for use in the second phase. The described method is flexible, does not require model retraining, and can be integrated into a variety of popular frameworks. The authors appear prepared to publicly release the code for their method.

**Audience:**

Yes

**Broader Impact Concerns:**

The authors have adequately addressed broader impact concerns in their statement in the paper.

**Claims And Evidence:**

Yes

**Requested Changes:**

The only absolute must in my opinion is to integrate some discussion of reference [1] into the paper. Another proofreading pass would be my next priority. The other things I mentioned above would be nice to have in the paper for completeness but should not be considered obligatory.

**Strengths And Weaknesses:**

### **Strengths**

My overall impression of the paper is positive. The authors have presented a largely self-contained, well-motivated, and well-presented piece of work. The exposition and notation are clear and easy to follow. The initial experimental analysis establishing the biphasic behavior of cross- and self-attention during diffusion sampling is compelling, and the proposed method is well motivated by it. The method itself is quite flexible and appears to require little overhead of its own, and it does not require any retraining of the underlying models. The experimental results demonstrating the method show that it is effective and support the paper's core claims. The authors apparently plan to publicly release their code.

### **Weaknesses**

* Most issues with the paper are quite minor and could be corrected with an additional round of proofreading (e.g. a few punctuation errors, including doubled commas in one spot; an error in the Jarzynski citation at the beginning of Section 2, which includes the year twice; spelling out `log` in math mode versus using `\log`, etc.).
* The only major issue that I see as glaring is that one of the paper's main findings, namely about the biphasic behavior of cross- and self-attention, is already known in the literature but not cited. By current count, this earlier work [1] already has 630 citations, so it is an unfortunate omission. However, what the authors *do* with this knowledge in the current submission is a worthy and (I believe) novel contribution in itself, so this is not a fatal error. Nevertheless, this earlier work should absolutely be acknowledged.

### **Additional Commentary**

There is some additional related work that may be worth discussing in the context of adaptive guidance. Reference [2] investigates the role of guidance during different stages of sampling and proposes applying guidance only within a limited interval. Reference [3] explores two strategies for applying guidance: The first weights the influence of guidance using a schedule, and the second anneals the condition itself with decreasing Gaussian noise during sampling. I would be interested in seeing the authors' perspective reconciling the findings reported in these papers with their analysis.

One of the biggest implicit questions all papers face is "Who cares?" I think the authors have done a good job acknowledging the limitations of their work by putting it in a global perspective. While an individual user may or may not notice a 10 percent reduction in computation cost, it is definitely something that adds up when one considers the sheer popularity of these models. I wanted to point out what I thought was a particularly good answer on the authors' part to this implied question.

What may be the bigger limitation is also acknowledged by the authors, namely that of the poorer performance of their method on text-image alignment. I think the authors have provided a reasonable response here by introducing more flexibility into the model, but it would be interesting to have a bit more insight into this issue.

### **References**

[1] Balaji, Yogesh, et al. "ediff-i: Text-to-image diffusion models with an ensemble of expert denoisers." arXiv preprint arXiv:2211.01324 (2022).

[2] Kynkäänniemi, Tuomas, et al. "Applying guidance in a limited interval improves sample and distribution quality in diffusion models." arXiv preprint arXiv:2404.07724 (2024).

[3] Sadat, Seyedmorteza, et al. "CADS: Unleashing the diversity of diffusion models through condition-annealed sampling." arXiv preprint arXiv:2310.17347 (2023).

---

> ### Author Response · Authors · 2025-01-21
> **Response to Reviewer bq4G**
>
> We appreciate the reviewer’s time and effort in evaluating our submission and are encouraged by the positive feedback. Below, we address each of the reviewer's comments in detail, aiming to further alleviate any concerns.
>
>  ### Proofreading the Manuscripts
> Thanks for the pointing out. We carefully proofread our manuscript, and list our updates as below:
>
> > a few punctuation errors, including doubled commas in one spot
>
> In our revised manuscript, we corrected instances such as `i.e.,, temporally gating` to `i.e., temporally gating`, `i.e.,, U-Net ` to `i.e., U-Net` and `i.e.,, DiT ` to `i.e., DiT`
>
> > an error in the Jarzynski citation at the beginning of Section 2
>
> We have updated the citation from `dating back to nonequilibrium statistical physics (1997) (Jarzynski, 1997)` to  `dating back to nonequilibrium statistical physics (Jarzynski, 1997)`
>
> > spelling out `log` in math mode versus using `\log`
>
> We have updated the manuscript and highlighted the changes in orange.
>
> ### Comparison with Related Works.
>
> Thanks for the pointing out. We have incorporated the related works into our revised manuscript, which is colored orange. We are encouraged by the reviewer's kind words like `what the authors do with this knowledge in the current submission is a worthy and (I believe) novel contribution in itself.` Additionally, following the reviewer's suggestion, we detailed the connections and differences between our study and these related works, which can help further clarify the position of this paper.
>
> > Comparison with eDIFF-I [1]
>
> 1) Both eDIFF-I and TGATE analyze the temporal behavior of the diffusion model using different methods but reach similar conclusions. While eDIFF-I presents attention maps and prompt switching during denoising, this study explores attention mechanisms by tracking activation changes, evaluating text embedding impacts, and using caching to explore different attention modules' influence. Methodologically, they are different.
> 2) Beyond the analytical setting, this paper further explores self-attention behavior across time steps, revealing a biphasic interaction between self- and cross-attention. While eDIFF-I also notes that later denoising steps prioritize quality over alignment, it does not link this observation to self-attention.
> 3) Compared to eDIFF-I's single qualitative samples, our study offers a systematic analysis using both quantitative and qualitative methods across diverse model architectures, noise schedulers, and evaluation benchmarks. Our comprehensive analysis, which is more numerical and reproducible, provides valuable insights and robust evidence to guide future exploration.
> 4) Beyond their distinct analyses of temporal dynamics, eDIFF-I, and TGATE target different objectives. eDIFF-I integrates ensemble learning into diffusion models by aligning model design with temporal dynamics. In contrast, TGATE focuses on improving diffusion model efficiency, which tries to duplicate the success of KV-cache (in LLM) to the diffusion model.
>
> We recognize the strong correlation between eDIFF-I and TGATE and have updated our related work to ensure proper credit.
>
> > Compared with other Adaptive Classifier Free Guidance.
>
> 1) These approaches design a step-specific strategy for classifier-free guidance. Our analysis provides strong evidence to further support their proposed methods.
> 2) While we acknowledge the connection between classifier-free guidance and cross-attention behavior, our study extends the analysis to self-attention, offering insights beyond the scope of adaptive guidance.
> 3) Moreover, our primary technical contribution lies in feature caching rather than eliminating classifier-free guidance. As demonstrated in Section 6.3, our method is compatible with adaptive guidance strategies, serving as an orthogonal solution to improve efficiency.
>
> We also include these studies in related works.
>
> ### Text-image alignment
>
> Following the reviewer’s suggestion, we provide visualization results in Appendix H. As shown in the figure, TGATE maintains global semantics well, even at high acceleration rates. However, higher rates can cause minor distortions, such as altering the bowl shape or breaking the bracelet. We argue these distortions may affect model detection scores, explaining the slight performance drop on the text-image alignment benchmark. For applications requiring high quality, we recommend a higher m and a smaller k acceleration. In less demanding cases, the trade-off is acceptable, given the achieved acceleration.

---

> > ### Comment · Reviewer_bq4G · 2025-01-22
> > **Response to Authors**
> >
> > I thank the authors for their thorough response to my remarks. All issues that I raised have been addressed to my satisfaction.
> >
> > As a minor note, I'll mention that a few instances of `log` versus `\log` are still hanging around in math mode, e.g. on page 3 in the text between equations (2) and (3). It's not a big deal, but I thought I'd point it out since it is correct elsewhere in the paper.

---

> > > ### Author Response · Authors · 2025-01-22
> > >
> > > We are pleased to hear that you are satisfied with our response.
> > > We also appreciate your valuable reminder and have incorporated the revisions in our updated manuscript.

---

### Review · Reviewer_vkDp · 2024-11-23

**Summary Of Contributions:**

This paper analyzes (empirically) the role of attention mechanisms in text-conditional diffusion models during the image generation process. The authors find that the cross-attention module is crucial in the initial *semantic planning* phase, where the model interprets the text prompt, while the self-attention module is more important in the later *fidelity-improving* phase, where the model generates the actual image details.  Based on this observation, the paper introduces a simple, training-free method called Temporally Gating the Attention (TGATE), which accelerates image generation by caching and reusing attention outputs at previous timesteps. In the fidelity-improving phase, the TGATE method caches and reuses the cross-attention output, while in the semantic-planning phase, self-attention predictions are cached and reused. The authors show that TGATE can accelerate various text-conditional diffusion models by 10-50% without significantly impacting the quality of the generated images.

**Audience:**

Yes

**Broader Impact Concerns:**

Limitations and Broader Impacts considerations are correctly discussed within the main body of the paper.

**Claims And Evidence:**

Yes

**Requested Changes:**

No major change is requested as the paper is well-written and clear.

A few things I believe will make this paper better:

- As mentioned, add more discussion about Self-Attention experiments in the main body of the paper.
- Introduce the sampler for the experiments as early as possible. Right now, in Sec 3.1, there's no information about the sampler, just that convergence of cross-attention appears between 5-10 steps. All this, should be relative to the total number of steps, and the actual algorithm used. Note that DPM solver is introduced in Sec 3.2.
- I would recommend adding more information about the Attention mechanism to make the paper more self-contained (Eq. 3). For example, you could give a little more information about K, Q, V.
- Table 1. "Generation Performance" --> "FID" or "Generation Performance (FID)"
- Page 9 - There's a line break in Appendix E (in latex: Appendix~E)
- References. Check consistency in the format. Also some references are duplicated: Schmidhuber 1992b = Schimidhuber 1992c.

**Strengths And Weaknesses:**

Strengths:
- Very interesting analysis of the role of Attention in text-to-image diffusion models.
- The paper introduces a simple method to exploit the observations to make diffusion sampling more efficient.
- Many comparisons to other acceleration methods, and also different results with different popular text-to-image diffusion models.
- The method provides a nice acceleration (10-50%) with minimal impact in quality.

Weaknesses:
- No major weaknesses.

Comments:
- Eq (6). Can you explain better how is that the average is used as anchor. I would expect that we need access to both of them (or one of them, but not the average).
- It would be good to add more details in the body of the paper about the Self Attention experiments. Right now, the sections analyzing self-attention just point to the Appendix (e.g., Sec 3.3 and Sec 4.2).
- It would be nice to see more discussion on the role of the sampler used in the Experiments (all the experiments and also the ones in Table 6).

---

> ### Author Response · Authors · 2025-01-21
> **Response to  Reviewer vkDp**
>
> We sincerely appreciate the reviewer’s constructive suggestions, comments, and encouragement. We have carefully addressed the concerns and incorporated the feedback to enhance our manuscript, as detailed below.
>
>  ### Explaining Eq. (6).
> We primarily follow the philosophy of classifier-free guidance, averaging the two branches in a weighted manner. As shown in Fig. 2, our pilot study indicates that cross-attention maps converge to a fixed point after a few inference steps, suggesting increasing similarity between unconditional and conditional maps over time. Thus, we hypothesize that using features from a single branch or their average has minimal impact. To validate this, we provide a qualitative analysis in Appendix K, showing that the final outputs from different branches are visually indistinguishable.
>
>  ### ADD self-attention analysis to Main Body.
> Thank you for the suggestion. We relocated the self-attention analysis to the main body, highlighting it in orange for clarity.
>
>
> ### Details for Noise Schedulers.
> We have updated the schedulers in Sections 3.1 and 6. Briefly, we use a DPM-solver with 25 inference steps throughout Section 3. In Section 6, unless otherwise specified, we adhere to the default noise scheduler and inference step settings for each model.
>
>  ### Polishing Paper.
> Thank you for the suggestion. We have expanded the introduction to the attention mechanism in Section 2, updated the column names in Table 1, fixed the line break in Appendix E, removed duplicate citations, and standardized the citation format.

---

> > ### Comment · Reviewer_vkDp · 2025-01-22
> >
> > Thanks for the response. The updated version of the manuscript looks good to me.

---

> > > ### Author Response · Authors · 2025-01-23
> > >
> > > Thank you for reviewing the updated manuscript and for your positive feedback. We are glad to hear it meets your expectations.

---

### Review · Reviewer_s3sc · 2025-01-16

**Summary Of Contributions:**

The paper introduces TGATE (Temporally Gating the Attention), a method for accelerating text-conditional diffusion models by analyzing the roles of cross-attention and self-attention across different phases of the inference process. The authors observe that cross-attention becomes less critical after an initial “semantic-planning phase,” while self-attention grows more influential in a subsequent “fidelity-improving phase.” Leveraging this insight, TGATE selectively caches and reuses attention outputs, reducing computational overhead without retraining the model.

The authors demonstrate the efficacy of TGATE across multiple diffusion models, showing computational efficiency gains (up to 50%) without requiring additional training. They provide comprehensive evaluations across datasets, models, and configurations, establishing TGATE as a versatile tool for accelerating diffusion-based generative tasks.

**Audience:**

Yes

**Broader Impact Concerns:**

N.A.

**Claims And Evidence:**

Yes

**Requested Changes:**

Please refer to the "Strengths and Weaknesses" section.

**Strengths And Weaknesses:**

Strengths:
1. The paper provides a comprehensive experimental analysis of the roles of cross-attention and self-attention at each denoising step in diffusion models. Leveraging these observations, the authors propose TGATE, a training-free acceleration method that achieves up to a 50% reduction in MACs and latency with minimal impact on generation quality.
2. TGATE is extensively validated across diverse architectures (e.g., Transformer and U-Net), tasks (e.g., text-to-image and text-to-video), and sampling schedulers. Additionally, it demonstrates the ability to slightly accelerate latent consistency models, further showcasing its versatility.


Major Weakness:

1. TGATE relies on the hyper-parameters gate step (m) and interval (k), which require careful tuning to achieve optimal performance across different use cases. Currently, k is often set to fixed values such as 10 or 15. A more systematic and flexible approach would be to relate k and m to the ratio of total sampling steps, allowing these parameters to scale with the specific inference settings. For example, the authors could provide a threshold based on the change in cross/self-attention differences to determine an appropriate ratio, thereby offering a more adaptive and generalizable framework for hyper-parameter selection.

2. Several widely-used open-source models, such as SD3, Flux and its successors in video generation (e.g., CogVideo X), are built on the multi-modal DiT (MM-DiT) framework. This architecture employs attention mechanisms that may differ from the cross-attention + self-attention structure analyzed in this paper. The authors could enhance the work by including a discussion on how TGATE applies to MM-DiT and similar multi-modal architectures.

Minor Weakness:

* In Section 3.2, the conditional text embedding $c$ is replaced with the null text embedding $\emptyset$, whereas in Section 4.1, the method caches and reuses the mean of the cross-attention maps $C_c$ and $C_{\emptyset}$. Considering this gap, It would be worthwhile to explore the difference of reusing only $C_{\emptyset}$ in fidelity-improving stage.
* The definition of Multiply-Accumulate Operations (MACs), mentioned in the third line under Figure 5, should be introduced earlier in the paper for clarity.
* Providing the prompts used to generate the main qualitative results would enhance reproducibility and help readers better understand the context of the examples.
* Including qualitative results for video generation would offer a more comprehensive evaluation, especially considering the distinct challenges associated with temporal consistency in video synthesis.
* There is a noticeable reduction in the properties evaluated by T2I-CompBench. To address this, additional qualitative results could be presented to illustrate the influence of TGATE on handling complex relationships, such as attribute-object interactions. This would provide deeper insights into TGATE’s impact on maintaining semantic consistency in challenging scenarios.

---

> ### Author Response · Authors · 2025-01-21
> **Response to reviewer s3sc**
>
> We sincerely appreciate the reviewer's comments. Below, we provide our response in detail.
>
>
> ### Relating k and m to the ratio of total sampling steps.
> Thank you for the suggestion. We agree that defining hyperparameters as a ratio of inference steps can enhance robustness. In response, we have added a new section (Appendix L) to discuss the hyperparameter settings and present empirical studies demonstrating that setting fixed ratios for `k` and `m` ensures stable performance across varying inference steps.
>
> ### TGATE and MM-DiT
> Thank you for the reviewer’s comment.
> We recognize the significance of MM-DiT and will prioritize exploring its applicability in our future work.
> As a potential solution, we may strategically cache features from both the text and image branches at periodic intervals across different time steps.
> To clarify our position regarding the emergence of MM-DiT, we outline the following points:
>
> 1) We agree that MM-DiT demonstrates strong results in scalable scenarios, such as text-to-image and text-to-video generation. However, it is worth noting that recent product-level models [A] often favor pure transformer architectures to align with infrastructure designs optimized for LLMs. Thus, we believe that investigating the behavior of pure transformer architectures remains practically beneficial.
> 2) TGATE is among the first studies to focus on the design of transformer-based feature caching. To maximize its impact, we adopted a pure transformer backbone. While we recognize the importance of applying TGATE to MM-DiT, we have listed this as a high-priority task in our future research plans.
>
>
> [A] Polyak, Adam, et al. "Movie gen: A cast of media foundation models." arXiv preprint arXiv:2410.13720 (2024).
>
> ### Different Anchors in the Fidelity-Improving Stage
> The reason behind using averaging features is to follow the CFG, combining the two branches through weighted averaging.  Fig. 2 shows that cross-attention maps stabilize to a fixed point after a few inference steps, indicating growing similarity between unconditional and conditional maps over time. Based on this observation, we hypothesize that using features from either branch or their average has negligible impact. To support this hypothesis, we update a qualitative analysis in Appendix K, demonstrating that the final outputs from different branches are visually indistinguishable.
>
> ### Introducing MACs earlier
> Thank you for pointing that out. We have moved the introduction of MACs to the beginning of the experimental section and highlighted it in orange for clarity.
>
> ### Providing the Used Prompts
>
> We included a table in Appendix M to present all prompts used in our paper, ensuring reproducibility.
>
> ### Additional Visualzations
> We update video generation results in Appendix I and the qualitative analysis based on T2I-Compbench prompts in Appendix H.
> As we introduced in our limitation discussion, higher rates may lead to some minor distortions, such as altering the bowl shape or breaking the bracelet (see visualizations). We suggest that these distortions could influence model detection scores, accounting for the slight performance decline on the text-image alignment benchmark.  Considering the acceleration from TGATE, this trade-off is acceptable in most cases that do not demand
> exceptionally high-quality images.

---

> > ### Comment · Reviewer_s3sc · 2025-01-23
> >
> > Thanks for the rebuttal. My concerns have been addressed.

---

> > > ### Author Response · Authors · 2025-01-23
> > >
> > > We are happy to hear that your concerns have been addressed. Thank you for your feedback and constructive discussion.

---

### Decision · Action_Editor_jS4S · 2025-02-18

**Recommendation:** Accept as is

**Comment:**

The decision is based on its strong empirical analysis, novel insights into cross- and self-attention roles in diffusion models, and the practical efficiency improvements introduced by TGATE. Reviewers praised its clear motivation, solid experimental validation across models and tasks, and minimal quality loss despite acceleration. Minor concerns about hyperparameter tuning and applicability to MM-DiT were addressed satisfactorily. The reviewers provided constructive feedback, which the authors incorporated in the revision.

**Audience:**

broad interests in diffusion/flow models.

**Claims And Evidence:**

The authors present a thorough empirical analysis of cross-attention and self-attention in diffusion models, using both quantitative metrics and qualitative visualizations. Their proposed TGATE method is validated across multiple architectures, tasks, and datasets, demonstrating consistent efficiency improvements (10-50%) with minimal quality loss. Reviewers acknowledge the robustness of the evidence, including comparisons with existing acceleration techniques. Minor concerns about hyperparameter tuning and broader applicability to certain architectures (e.g., MM-DiT) were addressed in the rebuttal. All reviewers suggest accept.